

# Atmospheric Carbonyl Sulphide (OCS) measured remotely by FTIR solar absorption spectrometry

Geoffrey C. Toon, Jean-Francois L. Blavier, Keeyoon Sung

Jet Propulsion Laboratory, California Institute of Technology

Correspondence to: Geoffrey.C.Toon@jpl.nasa.gov

**Abstract.**  Atmospheric OCS abundances have been retrieved from spectra measured by the JPL

MkIV Fourier Transform Infra-Red (FTIR) spectrometer during 24 balloon flights and during nearly 1100 days of ground-based observations since 1985.  Our spectral fitting approach uses broad windows to enhance the precision and robustness of the retrievals.  Since OCS has a vertical profile similar in shape to that of $N_2O$, and since tropospheric $N_2O$ is very stable, we reference the OCS observations to those of $N_2O$, measured simultaneously in the same airmass, to

remove the effects of stratospheric transport, allowing a clearer assessment of secular changes in OCS.  Balloon measurements reveal less than 5% change in stratospheric OCS amounts over the past 25 years.  In the troposphere a springtime peak of tropospheric OCS is seen, followed by a rapid early summer decrease, similar to the behavior of $CO_2$.  This results in a peak-to-peak seasonal cycle of 5-6% of the total OCS column at Northern mid-latitudes.  In the long-term

tropospheric OCS record, a 5% decrease is seen during 1990-2002, followed by a 5% increase from 2003 to 2012.

## Introduction

With a column-averaged mole fraction of 450 ppt, OCS is the most abundant sulphur-containing gas in the atmosphere, except following major volcanic eruptions when $SO_2$ briefly

dominates.  OCS sources are at the surface: biogenic ocean activity produces OCS directly and also indirectly via oxidation of di-methyl-sulphide (DMS) and $CS_2$ [Kettle et al., 2002].  On land, the rayon industry emits carbon disulphide ($CS_2$), and there is evidence for biomass burning producing OCS which is lofted into the upper troposphere [Notholt et al., 2003].

The sinks of OCS are uptake by vegetation and soils, and at higher altitudes OCS is

destroyed by OH and photolysis, leading to the formation of $SO_2$, which becomes a major source of non-volcanic SSA (stratospheric sulphate aerosol) [Crutzen, 1976; Wilson et al., 2008].  There is no consensus on the proportion of SSA that result from OCS versus direct volcanic injection of $SO_2$ (Leung et al., 2002).  The SSA has an important impact on the radiation budget, transport,





and chemistry [Crutzen, 1976].  Its large surface area catalyzes heterogeneous reactions, some of

which affect the stratospheric $O_3$ layer.

The ingestion of OCS by plants, similar to $CO_2$, is a diagnostic of the carbon cycle.  The impact of biological activity on OCS abundance in the lower troposphere is (in fractional terms) much larger than $CO_2$.  For example, the plant-induced seasonal cycle of $CO_2$ is about 2% of the atmospheric column at northern mid-latitudes, whereas for OCS it is closer to 10%, and for

tropospheric OCS it is ~15% [Campbell et al., 2008; 2015, Dlugokencky et al., 2001; Montzka et al., 2007].  OCS is shorter-lived than $CO_2$ and with smaller sources, with the result that its atmospheric column is a million times smaller.  The precision and accuracy of OCS measurements is therefore much poorer than those of $CO_2$.

OCS was first measured from space by the ATMOS solar occultation FTS in 1985

(Zander et al., 1988).  Since then the ACE instrument has reported OCS measurements (Rinsland et al., 2007; Barkley et al. 2008).  More recently, Glatthor et al., [2014; 2015] reported OCS profiles retrieved from spectra measured by the MIPAS instrument on board the ENVISAAT satellite.  Kuai et al. [2014] reported OCS column measurements from the TES instruments on board the Aura satellite.  Vincent and Dudhia [2016] reported OCS measurements from the IASI

instrument.

Remote measurements of OCS from space promise new insights on the carbon cycle, provided that other factors that govern column OCS (e.g., stratospheric transport) can be correctly accounted for.  The fact that OCS is shorter lived than $CO_2$ means that its vmr profile decreases more rapidly with altitude in the stratosphere.  Thus vertical transport in the stratosphere has far

more impact on the total column amounts of OCS than $CO_2$, complicating attempts to accurately determine tropospheric OCS amounts from space.  For example, if the OCS mole fraction were constant in the troposphere and decrease linearly with pressure above the tropopause, then a change in the tropopause altitude from 300 to 200 mbar (9 to 12 km) would result in a 5% increase in the total OCS column OCS column above sea-level.  This is likely larger than the

tropospheric variations of interest, e.g. due to exchange with the surface, especially in the southern hemisphere where there is little land at mid-latitudes.  So accounting/correcting for these stratospheric transport effects is an essential prerequisite to gaining insights into tropospheric OCS variations from total column measurements.

We show in this paper that the OCS and $N_2O$ vmr profiles are similar in shape in the

stratosphere and are strongly correlated, both being affected similarly by transport.  Since the tropospheric $N_2O$ amount varies very little, the $N_2O$ column can be used to account for transport-



driven changes in the stratospheric OCS amount, allowing the tropospheric OCS behavior to be more clearly seen.

## MkIV Instrument

The MkIV FTS is a double-passed FTIR spectrometer designed and built at JPL in 1984 for atmospheric observations [Toon, 1991]. It covers the entire 650-5650 cm$^{-1}$ region simultaneously with two detectors: a HgCdTe photoconductor covering 650-1800 cm$^{-1}$ and an InSb photodiode covering 1800-5650 cm$^{-1}$. The MkIV instrument has flown 24 balloon flights since 1989. It has also flown on over 40 flights of the NASA DC-8 aircraft as part of various

campaigns during 1987 to 1992 studying high-latitude ozone loss. MkIV has also made 1090 days of ground-based observations since 1985 from a dozen different sites, from Antarctica to the Arctic, from sea-level to 3.8 km altitude. MkIV observations have been extensively compared with satellite remote sounders (e.g. Velazco et al. 2011) and with in situ data (e.g., Toon et al., 1999).

## Analysis Methods


        The spectral fitting was performed with the Version 4.8 GFIT (Gas Fitting) code, a non-linear least-squares algorithm developed at JPL. GFIT scales user-prescribed a priori atmospheric gas vmr profiles to fit calculated spectra to those measured. For balloon observations, the atmosphere was discretized into 100 layers of 1 km thickness. For ground-

based observations, 70 layers of 1 km thickness were used. Absorption coefficients were computed line-by-line assuming a Voigt lineshape and using the ATM linelist [Toon, 2014a] for the telluric lines. This is a "greatest hits" compilation, founded on HITRAN, but not always the latest version for every band of every gas. In situations where the latest HITRAN version (Rothman et al., 2013) gave poorer fits, the earlier HITRAN version was retained. For OCS the

ATM linelist is based on HITRAN 2012, but with 709 additional lines, empirically determined from laboratory spectra, representing missing hot-bands of the main isotopolog. The solar linelist [Toon, 2014b] used in the analysis of the ground-based spectra was obtained from analysis of low-airmass spectra observed during balloon flights of the MkIV and shuttle flights of the ATMOS instruments.

Sen et al. [1996] provide a more detailed description of the use of the GFIT code for retrieval of vmr profiles from MkIV balloon spectra. GFIT was previously used for the Version 3 analysis [Irion et al., 2002] of spectra measured by the Atmospheric Trace Molecule Occultation



Spectrometer (ATMOS), and is currently used for analysis of TCCON spectra [Wunch et al., 2011] and MkIV spectra [Toon, 2016].

**Balloon Observations**

The MkIV instrument has made 24 balloon flights since 1989. Several of these provided multiple occultations (e.g., sunset and sunrise during same flight), so that in total we have 30 profiles, which are summarized in Table 1. The flights are predominantly from around 35N, except for the 1997-2002 period when several high latitude flights were undertaken from

Fairbanks, Alaska, and Esrange, Sweden.

| Date | Lat deg. | Long deg. | $Z_{min}$ km | $Z_{max}$ km | Event | MCF | Launch Site Town | Launch Site State |
|------|------|------|------|------|-------|-----|------|------|
| 05-Oct-1989 | 34.6 | -105.3 | 13 | 37 | Sunset | 28 | Ft. Sumner | New Mexico |
| 27-Sep-1990 | 34.2 | -105.6 | 10 | 36 | Sunset | 28 | Ft. Sumner | New Mexico |
| 05-May-1991 | 37.5 | -111.5 | 15 | 37 | Sunset | 28 | Ft. Sumner | New Mexico |
| 06-May-1991 | 36.5 | -113.0 | 15 | 32 | Sunrise | | | |
| 14-Sep-1992 | 35.2 | -110.9 | 23 | 39 | Sunset | 28 | Ft. Sumner | New Mexico |
| 15-Sep-1992 | 35.3 | -104.0 | 22 | 41 | Sunrise | | | |
| 03-Apr-1993 | 34.8 | -114.8 | 17 | 37 | Sunset | 28 | Daggett | California |
| 25-Sep-1993 | 34.0 | -107.5 | 6 | 38 | Sunset | 28 | Ft. Sumner | New Mexico |
| 26-Sep-1993 | 33.1 | -95.3 | 13 | 38 | Sunrise | | | |
| 22-May-1994 | 36.1 | -108.6 | 14 | 36 | Sunset | 28 | Ft. Sumner | New Mexico |
| 23-May-1994 | 36.3 | -100.9 | 11 | 37 | Sunrise | | | |
| 24-Jul-1996 | 56.7 | -100.9 | 11 | 24 | Ascent[*] | 28 | Lynn Lake | Manitoba |
| 28-Sep-1996 | 32.7 | -113.1 | 4 | 38 | Sunset | 28 | Ft. Sumner | New Mexico |
| 08-May-1997 | 68.7 | -146.0 | 8 | 38 | Sunrise | 24 | Fairbanks | Alaska |
| 08-Jul-1997 | 66.4 | -148.3 | 7 | 32 | Ascent | 11 | Fairbanks | Alaska |
| 08-Jul-1997 | 64.7 | -150.2 | 9 | 32 | Descent | | | |
| 03-Dec-1999 | 64.2 | 19.3 | 6 | 34 | Sunset | 11 | Esrange | Sweden |
| 15-Mar-2000 | 67.8 | 34.2 | 11 | 29 | Sunrise | 4 | Esrange | Sweden |
| 16-Dec-2002 | 64.4 | 31.2 | 12 | 31 | Sunrise | 5 | Esrange | Sweden |
| 01-Apr-2003 | 68.3 | 35.2 | 11 | 32 | Sunrise | 5 | Esrange | Sweden |
| 19-Sep-2003 | 34.3 | -113.3 | 7 | 36 | Sunset | 28 | Ft. Sumner | New Mexico |
| 23-Sep-2004 | 33.8 | -109.2 | 11 | 38 | Sunset | 28 | Ft. Sumner | New Mexico |
| 20-Sep-2005 | 35.2 | -114.1 | 11 | 39 | Sunset | 39 | Ft. Sumner | New Mexico |
| 21-Sep-2005 | 34.0 | -110.3 | 13 | 29 | Sunrise | | | |
| 07-Feb-2007 | 67.9 | 21.0 | N/A | 34 | Ascent[#] | 11 | Esrange | Sweden |
| 22-Feb-2007 | 67.9 | 21.1 | 25 | 34 | Ascent[#] | 11 | Esrange | Sweden |
| 22-Sep-2007 | 35.2 | -114.1 | 10 | 38 | Sunset | 28 | Ft. Sumner | New Mexico |
| 23-Sep-2007 | 34.0 | -110.3 | 13 | 38 | Sunrise | | | |
| 23-Sep-2011 | 34.5 | -108.8 | 6 | 39 | Sunset | 28 | Ft. Sumner | New Mexico |
| 24-Sep-2011 | 35.7 | -96.3 | 14 | 40 | Sunrise | 28 | | |
| 13-Sep-2014 | 36.2 | -112.5 | 7 | 39 | Sunset | 28 | Ft. Sumner | New Mexico |
| 14-Sep-2014 | 35.6 | -103.5 | 8 | 40 | Sunrise | 28 | | |
| 27-Sep-2016 | 36.0 | -110.5 | 11 | 39 | Sunset | 28 | Ft. Sumner | New Mexico |

**Table 1.** *Summary of MkIV balloon occultations. Lat and Long represent the latitude and longitudes of the 20 km tangent point. $Z_{min}$ and $Z_{max}$ represent the altitudes over which the*





*tangent altitude varied. MCF represents the size of the balloon in Millions of Cubic Feet.*

*Flights flagged by [#] or [*] acquired no useful data.*


We measured OCS using 5 different windows (see Table 2). The windows at 2050 and 2069 cm$^{-1}$ cover the P- and R-branches of the $v_3$ OCS band, which is 80 times stronger than any other OCS band. These $v_3$ windows provide nearly all the stratospheric OCS information, but at lower altitudes these windows become increasingly blacked out due to $CO_2$, $H_2O$, and CO

absorption, as can be seen in the right-hand panel of Figure 1.

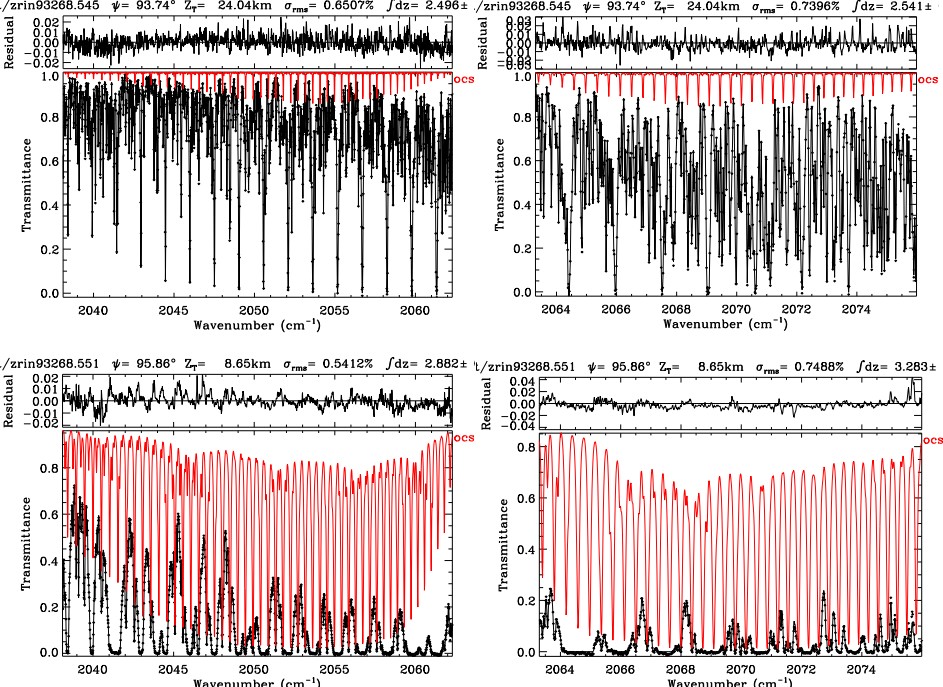

**Figure 1.** *Examples of spectral fits to MkIV balloon spectra measured at 24.0 km (top) and 8.7 km (bottom) tangent altitude. The left-hand panels show fits to the P-branch. Right panels show*

*fits to the R-branch. Black points represent the measured spectra. The black line is the fitted calculation. The red line shows the OCS contribution to the fitted calculation. Interfering gases are primarily $CO_2$, $H_2O$ and $O_3$. Below 10 km these windows become increasingly blacked out.*

Three additional windows, centered at 868, 2916, and 4096 cm$^{-1}$, containing much weaker OCS

absorption bands are therefore used to provide additional information at lower altitudes, and to provide a cross-check on the absolute OCS values retrieved in the $v_3$ band. It is important that the



strong and weak windows are consistent in terms of the retrieved OCS amounts, or else the retrieved vmr profiles will be skewed, with the $v_3$ band dominating above 10 km altitude, and the weaker bands contributing below.


| Center (cm$^{-1}$) | Width (cm$^{-1}$) | OCS Band | Interfering Gases Fitted | Retrieved Bias Factor | $S_{MAX}$ |
|---|---|---|---|---|---|
| 868.05 | 11.3 | $v_1$ | hno3  h2o  co2 | 1.115±0.097 | 1.54x10$^{-20}$ |
| 2050.20 | 24.2 | $v_3$ | co2  o3  h2o  co | 0.996±0.012 | 1.18x10$^{-18}$ |
| 2069.65 | 12.7 | $v_3$ | co2  o3  h2o  co | 0.996±0.019 | 1.25x10$^{-18}$ |
| 2915.55 | 38.0 | $v_1+v_3$ | ch4  o3  h2o  hcl no2  c2h6  hdo | 0.80±0.23 | 1.53x10$^{-20}$ |
| 4096.00 | 39.8 | $2v_3$ | ch4  h2o  hdo  hf | 0.95±0.24 | 8.26x10$^{-21}$ |

***Table 2.*** *Attributes of the five spectral windows used to fit OCS in MkIV balloon spectra. Center and Width denote the center wavenumber and width of the window. Band denotes the vibration-rotation state to which the OCS molecules were excited. $S_{MAX}$ is the maximum line intensity in units of cm$^{-1}$/(molec.cm$^{-2}$). The windows centered at 2050 and 2070 cm$^{-1}$ cover the P- and R-*

*branches of the strong $v_3$ band, and therefore allow accurate estimation of OCS amounts at the higher altitudes. The three weaker windows (868, 2916, and 4096 cm$^{-1}$) have larger biases and uncertainties, but are collectively consistent with respect to the two strong $v_3$ windows.*

Collectively, the three weak windows have an average bias close to 1.0, and therefore

won't skew the retrieved vmr profiles when results from the five windows are combined. Since the balloon spectra are ratioed (limb spectra divided by a high-sun spectrum), only 2 or 3 continuum basis functions are used, even for the widest windows. And no solar spectrum is needed.

The 41 $N_2O$ windows used in the analysis of MkV balloon spectra are listed at

http://mark4sun.jpl.nasa.gov/m4data.html. These cover the 1183 to 4750 cm$^{-1}$ spectral regions with widths up to 50 cm$^{-1}$. These include weak lines to accurately retrieve tropospheric $N_2O$ and strong lines for upper stratospheric $N_2O$.





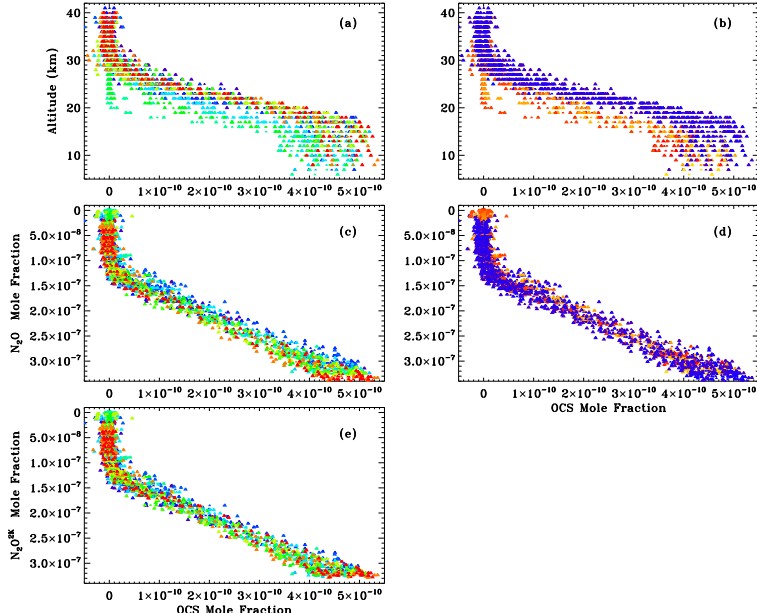

**Figure 2.** *Profiles of OCS retrieved from MkIV balloon spectra by averaging results obtained from the five windows listed in Table 1. In the left-hand panels the points are color-coded by year (e.g., blue=1990; green=2000, red=2014). In the right hand panels the exact same data are color-coded by latitude (blue=35N; red=67N). In the top panels, (a) and (b), the OCS is plotted versus altitude. In the middle panels, (c) and (d), the OCS is plotted versus $N_2O$. In the bottom panel (e) the $N_2O$ has been de-trended by 0.25%/year such that it represents the year 2000 ($N_2O^{2K}$). Only data with OCS uncertainties < 50 ppt and $N_2O$ uncertainties < 20 ppb were plotted, which reduced the total number of points in each panel from 756 to 668.*

Figure 2a shows 26 OCS profiles plotted versus altitude and color-coded by year (blue=1990; green=2000, red=2015). It is clear that the green points have lower OCS amounts than the other flights. Figure 2b shows the same data but color-coded by latitude (blue=35N; red=67N). The profiles measured at high latitude (Fairbanks in 1997; Esrange in 1999-2003) have much lower stratospheric OCS than the mid-latitude flights made prior and later. This is due to stratospheric descent at high latitude, especially in the winter.

Fortunately, the effects of transport, and their variations with latitude and season, can be largely be removed by using $N_2O$ as the vertical ordinate, as seen in the panels (c) and (d). This results in a much tighter consistency between the various profiles. It is clear from panel 2c that in the later years (red) there is more $N_2O$ at a given OCS value than in the early years (blue). Conversely, there is less OCS at a given $N_2O$ level. This implies an increasing trend in $N_2O$ or a



decreasing trend in OCS, or both. Panel 2d reveals that the OCS-$N_2O$ relationship is virtually
       independent of the measurement latitude, to within the scope and precision of these
       measurements. Incidentally, panel 2d shows that despite the mid-latitude balloon flights (blue)
       reaching higher altitudes (40 km) than the polar flights (red; 30-34 km), $N_2O$ never falls below 15
       ppb at mid-latitudes, whereas at high latitudes it falls to zero. This is a consequence of downward
transport in the stratosphere at high latitudes.

       Figure 2e shows the same data as in 2c, but with the $N_2O$ values adjusted for the known
       0.25%/year increase seen in in situ measurements. Over the timespan of the MkIV
       measurements, $N_2O$ has increased from 307.5 ppb in 1989 to 326.7 ppb in 2014, according to
       accurate in situ measurements (e.g., https://www.eea.europa.eu/data-and-
maps/daviz/atmospheric-concentration-of-carbon-dioxide-2#tab-chart_4). That is 6.24% in 25
       years or 0.25%/year. A similar rate of increase is to be expected in the stratosphere. It is
       fortunate that the increase has been so linear because this makes it independent of the age of the
       air, and hence altitude, simplifying the implementation of a correction.

       Correcting the measured $N_2O$ to its 2000 value eliminates the time-dependent creep in the
OCS-$N_2O$ relationship seen in fig 2c, resulting in a further tightening of their correlation. The
       OCS-$N_2O^{2K}$ relationship plotted in Fig. 2e is fairly linear for $N_2O^{2K}$ values down to 120 ppb,
       which represents ~10 mbar pressure or ~30 km at mid-altitudes. At higher altitudes the OCS goes
       to zero before $N_2O$, reflecting the shorter stratospheric lifetime of OCS. This causes a "knee" in
       the OCS-$N_2O$ relationship, such that the overall relationship can be reasonably approximated by
the equation

$$OCS = 2.25 \times 10^{-3} (N_2O^{2K} - 120 \text{ ppb}) \qquad N_2O^{2K} > 120 \text{ ppb}$$
$$OCS = \qquad\qquad 0 \qquad\qquad\qquad N_2O^{2K} < 120 \text{ ppb}$$

This relationship will later be used in the analysis of ground-based measurements.



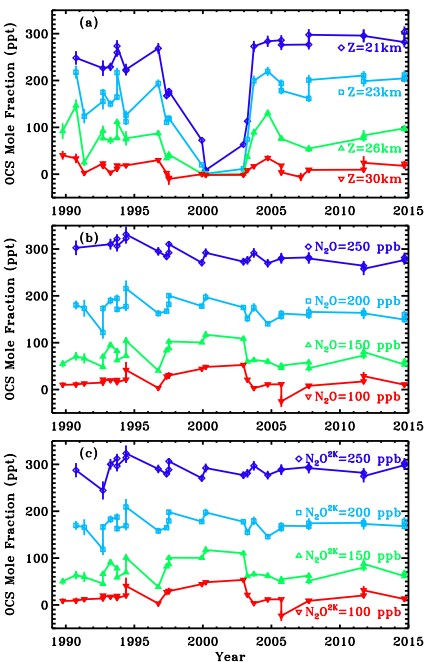

**Figure 3.** *(a) Stratospheric OCS dry mole fractions interpolated onto four different altitudes (21, 23 26, and 30 km) using the data from Fig.1. (b) The same OCS data interpolated to various $N_2O$ isopleths. The 250 ppb isopleth (red) corresponds to ~21 km altitude at mid-latitudes, whereas the 100 ppb isopleth (blue) corresponds to ~30 km. (c): Same as middle panel, but with the $N_2O$ amounts de-trended by 0.25%/year, as in fig.2e.*

Figure 3 shows the same OCS and $N_2O$ balloon data that was presented earlier in Fig. 2, but the OCS has been interpolated onto fixed altitudes and $N_2O$ isopleths. In Fig. 3a, the small OCS amounts from 1997 to 2002 were due to the balloon flights being undertaken at high latitudes. Plotted this way, these OCS data are clearly not useful for determining trend information. In fig. 3b, the same OCS data are interpolated onto various $N_2O$ isopleths. This removes transport-driven variations in the amounts of stratospheric OCS due to latitude or seasonal differences between flights, since these are common to OCS and $N_2O$. The resulting OCS appears to be decreasing with time at the lower altitudes (larger $N_2O$ isopleths), but this is an artifact of the increase of atmospheric $N_2O$.

As a result, the $N_2O$ isopleths get higher in altitude over time, which results in the appearance of a decreasing trend in an unchanging gas such as OCS. Fig 3c shows the same data as 3b, but with the $N_2O$ isopleths corrected to their values in the year 2000 under the assumption of a +0.25%/year trend. This eliminates artifacts due to the secular increase of $N_2O$, while still



preserving its ability to remove dynamically-induced fluctuations from the OCS record. Fig.3c

shows no significant trend in stratospheric OCS at any level. Based on this, we conclude that

stratospheric OCS hasn't changed by more then 5% over the past 25 years.

**Ground-based Observations**

The MkIV instrument has also made ground-based observations since 1985 from a dozen

different sites. Table 3 lists these sites, their locations, and the number of observations (Nobs) and

observation days (Nday) from each. The majority of the data come from JPL (0.35 km) and Mt.

Barcroft (3.8 km), both in California. On a typical day we might take 30 spectra at 120 cm Max

OPD over a period of 1.5 hours. After discarding bad spectra (e.g., clouds) the remainder are

averaged into forward-reverse pairs when the airmass is changing rapidly, or in fours or sixes at

lower airmasses. The net result is a few average spectra per day. We then use the GFIT algorithm

to retrieve vertical column abundances from these average spectra. Over 30 different gases are

retrieved, including OCS and $N_2O$. These results can be found at

http://mark4sun.jpl.nasa.gov/ground.html

| Town | State | Nobs | Nday | Latitude (deg.) | Longitude (deg.) | Altitude (km) |
|---|---|---|---|---|---|---|
| Esrange | Sweden | 160 | 32 | 67.889 | +21.085 | 0.271 |
| Fairbanks | Alaska | 124 | 46 | 64.830 | -147.614 | 0.182 |
| Manitoba | Canada | 20 | 5 | 56.858 | -101.066 | 0.354 |
| Mt. Barcroft | California | 1369 | 255 | 37.584 | -118.235 | 3.801 |
| Mtn View | California | 7 | 4 | 37.430 | -122.080 | 0.010 |
| Daggett | California | 33 | 21 | 34.856 | -116.790 | 0.626 |
| Ft Sumner | New Mexico | 216 | 71 | 34.480 | -104.220 | 1.260 |
| Wrightwood | California | 475 | 45 | 34.382 | -117.678 | 2.257 |
| JPL (B183) | California | 1709 | 577 | 34.199 | -118.174 | 0.345 |
| JPL (mesa) | California | 20 | 5 | 34.205 | -118.171 | 0.460 |
| Palestine | Texas | 4 | 3 | 31.780 | -95.700 | 0.100 |
| McMurdo | Antarctica | 37 | 20 | -77.847 | +166.728 | 0.100 |

*Table 3. MkIV ground-based observation sites, their locations and altitudes, and the*
*number of observations and observation days from each site as of the end of 2016, sorted*
*by latitude. Nobs is the number of observations. Nday is the number of observation days.*

We do not attempt to fit the whole spectrum. Instead we seek spectral regions in which

the absorption lines of the gases of interest (i.e. OCS, $N_2O$) are strong (but not saturated),

reasonably temperature-insensitive, and not overlapped by large residuals originating from

interfering gases (e.g. $H_2O$, CO, $CO_2$). Initially, 21 candidate OCS windows were defined and

analyzed in ground-based spectra, all from the strong $v_3$ band (see Table B.1). None of the



weaker OCS bands (at 868, 2915 and 4096 cm$^{-1}$) were used in the analysis of ground-based spectra: their OCS absorptions are simply too weak and overlapped with interfering absorptions.

Most of these 21 windows were taken from previous publications on ground-based OCS measurements: Griffith et al. [1998], Rinsland et al. [2002], Kryzstofiak et al. [2015], Kremser et al. [2015], and Lejeune et al., [2017]. In addition, four new, much-broader windows were also evaluated, in which most of the OCS lines are overlapped by stronger interfering absorbers. This would disqualify them in the traditional NDACC window selection process, which avoided strong

interferences. In the present work, however, the presence of an interfering absorption line overlapping the OCS line of interest isn't necessarily a disqualification, unless it produces a large residual.

To avoid a major digression, the details of the OCS window selection process are relegated to Appendix B. Suffice it to state here that just two OCS windows were eventually

chosen for subsequent use: a 13 cm$^{-1}$ wide window centered at 2051.3 cm$^{-1}$ containing 28 OCS P-branch lines, and a 9 cm$^{-1}$ wide window centered at 2071.1 cm$^{-1}$ containing 26 R-branch OCS lines. OCS amounts presented subsequently are the result of averaging these two windows.

A similar analysis was performed for the $N_2O$ windows used in this study. Since the tropospheric variations in $N_2O$ are small, the column variations are controlled mainly by

stratospheric transport. This allows the retrieved $N_2O$ amounts to be used to compensate for variations in the column OCS arising from stratospheric transport. The detailed discussion of the $N_2O$ window selection is relegated to Appendix C to avoid a major detour here. Suffice it to say that a subset of $N_2O$ windows was selected that gave high precision and consistency with averaging kernels similar to those of OCS. The latter was achieved by choosing windows with

$N_2O$ absorption depths similar to OCS, and discarding windows with strong $N_2O$ lines. Figure 4 compares averaging kernels for OCS and $N_2O$. Their strong similarity means that the information about atmospheric OCS and $N_2O$ has the same altitude distribution and is therefore directly comparable.






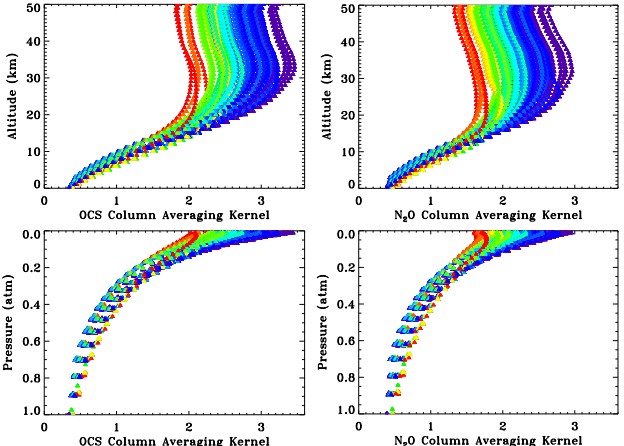

***Figure 4.*** *Ground-based column averaging kernels plotted versus altitude (upper panels) and pressure (lower panels). Left-hand panels show OCS kernels. Right-hand panels show $N_2O$ kernels. Values are color-coded by airmass (Purple=1; Red=10). A representative sub-set of 135 different observations, representing different sites and conditions, were used to make this*

*plot. The fact that the OCS and $N_2O$ kernels are similar in shape is not a fortuitous accident. The $N_2O$ windows used in this study were selected to contain weak $N_2O$ lines only, matching the OCS lines in terms of line depth and hence kernel shape. Since the shapes of the OCS and $N_2O$ kernels are similar, so will be their sensitivity to stratospheric transport.*

Figure 5 shows examples of fits to ground-based MkIV spectra in the two selected windows. The left panels show fits over the 2051 cm$^{-1}$ window covering 28 of the strongest P-branch lines of the $v_3$ band. The right panels show fits to the 2071 cm$^{-1}$ window covering 26 strong OCS lines in the center of the R-branch. The upper panels show fits to a low-airmass (SZA=23°) spectrum and the lower panels to a higher airmass spectrum (SZA=67°). Although

the OCS lines are stronger in the high aimass spectrum, so are the interfering absorptions. This tends to decrease the precision and accuracy of the OCS retrievals as zenith angle increases.

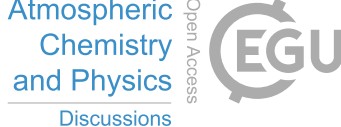

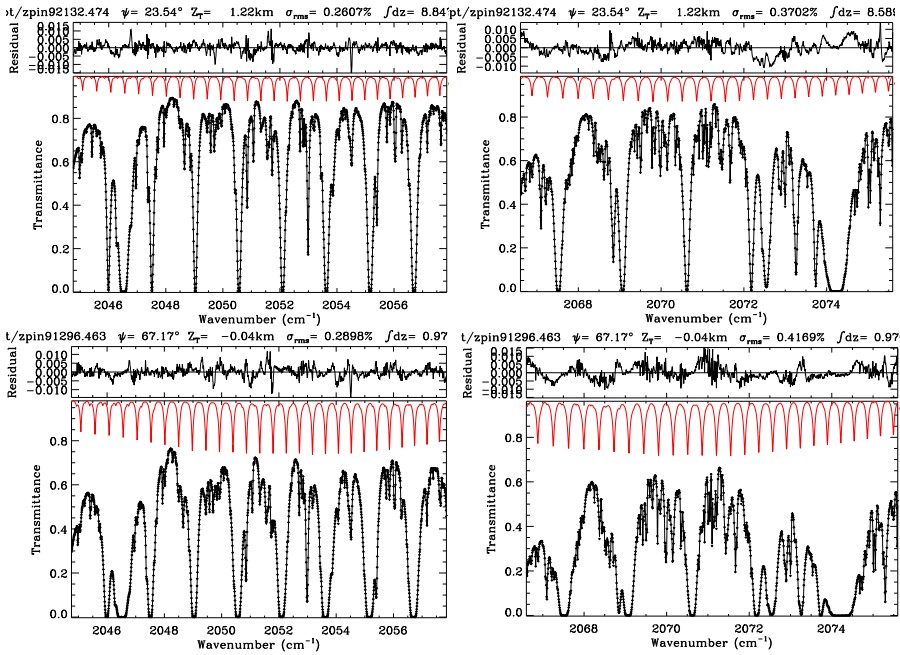


**Figure 5.** *Examples of spectral fits to ground-based MkIV spectra. The left panels show fits with the 2051 cm$^{-1}$ window covering most of the P-branch of the $v_3$ band. The right panels show fits to the 2071 cm$^{-1}$ window covering the center of the R-branch. The upper panels show fits to a low-airmass (SZA=23°) spectrum. The lower panels show fits to a higher airmass spectrum*

*(SZA=67°). Black symbols show the measured spectrum, and the black line is the fitted calculation. The red line is the OCS contribution to the fitted calculation. The individual contributions of other gases (mainly $H_2O$, $CO_2$, CO, $O_3$) are not shown because they would clutter the figure, obscuring the OCS. The saturated lines with a spacing of 1.6 cm$^{-1}$ are $CO_2$. Despite $CO_2$ being the strongest absorber, the residuals (Measured-Calculated) shown at the top*

*of each panel and are dominated by $H_2O$ and CO interferences.*

Figure 6a and 6b show OCS columns amounts retrieved from spectral fits such as those shown in Fig. 5. The points are color-coded by the logarithm of site altitude (blue=0 km; red=3.8 km). The same data are plotted in both panels, on the left versus year and on the right versus day

of year. Of course, the high altitude observations clearly show substantially less column OCS. Figs. 6c and 6d shows xOCS: the OCS column divided by the dry air column, the latter inferred from the measured surface pressure and the $H_2O$ column. This division improves the consistency of xOCS values retrieved from different altitude sites, but there is still a ~15% spread, which make it difficult to quantify the long-term trends and the seasonal cycle. Figs 6e and 6f show the

OCS/$N_2O$ column ratio. We know from the balloon measurements that OCS and $N_2O$ have





similarly-shaped vmr profiles (at least up to 30 km) and are therefore both subject to the same dynamical perturbations. So dividing the OCS by $N_2O$ removes most of the transport-driven geophysical noise. Thus the OCS/$N_2O$ column ratio is less variable than the xOCS, with probably a 10% spread of values, despite the $N_2O$ bringing noise into the ratio. However, since

the OCS/$N_2O$ relationship seen in the balloon data does not go linearly through the origin, the effects of stratospheric transport are greater on OCS than $N_2O$ and therefore do not completely cancel when taking the ratio.

To more completely remove stratospheric transport effects from the OCS measurements, we exploited the relationship established in Fig. 2e using MkIV balloon measurements.

Appendix A details how, under the assumption that the changes in $N_2O$ are entirely stratospheric in origin, the $N_2O$ column may be used to remove stratospheric transport effects from the OCS column measurements. Figures 6g and 6h show the ΔOCS as defined by equation A.3. This results in a slightly improved consistency between measurements made at different sites and a more compact seasonal cycle, as compared with figures 6e and 6f. The long-term changes (LTC)

of OCS are unfortunately still obscured by the seasonal cycle, which is roughly the same amplitude. Similarly, the seasonal cycle of OCS is obscured by the LTC. Due to the irregular sampling of the observations, the seasonal cycle can't simply be averaged out by smoothing the observations. For example, in some years we take data in Ft. Sumner, NM, in September only, near the minimum of the OCS seasonal cycle. This would drag down the mean for those years, if

the season cycle were not accounted for.

We therefore simultaneously fitted a linear spline and harmonic terms through the OCS data in figs 6g and 6h. The resulting function is continuous with respect to time. The seasonal cycle is fitted as independent sine and cosine terms with 12-, 6-, and 4-month periods (i.e., the first three harmonics), requiring 6 unknown parameters in total. The choice of 3 harmonics

reflects the lack of improvement in fitting the data with 4 harmonics. This parameterization assumes the seasonal variation is assumed to be identical each year and independent of the site (altitude or latitude).

The spline had knots at the beginning of each year, requiring 33 of these to cover 1985 to 2017. Between knots, the spline was assumed to be linear with time. The resulting matrix

equation is linear in the 33+6=39 unknown parameters, so no iteration is required. So 39 pieces of information are extracted from the 683 OCS measurements.

The use of a spline to represent the long-term changes has the advantage that an outlier in one particular year only affect the knots that bracket it. This is in contrast to fitting a polynomial (i.e., $y=a+bt+ct^2+dt^3+....$) which would allow an anomalous point at one end of the time series to





impact the fitted curve everywhere, and especially at the other end.  Figs. 6i and 6j show the LTC
         and the Seasonal Cycle (SC) extracted in this manner from the ΔOCS data.  Fig. 6i shows the
         spline values at the yearly knots, along with their uncertainties.  In years with little or no data
         (e.g. 1990, 2009) the spline values have large uncertainties.  Also, in years when the de-
         seasonalized xOCS amounts deviate from a straight line, uncertainties will be large.  The results

show a 5% drop (from 0.13 to 0.08) over the 1990 to 2002 period, followed by a 5% increase
         from 2002 to 2012.  Since 2012 ΔOCS has been flat.

         We note that the anomalously low data points in October 1986 were measured from
         McMurdo, Antarctica.  Since OCS was measured to be 10±4 % larger in the Arctic then the
         Antarctic (Notholt et al., 1997), the McMurdo measurements have much lower values than the

other points around that time, which were all in the Northern hemisphere.  These McMurdo
         points drag down the spline value at the 1987.0 knot in figure 6i.

         Figure 6j shows the seasonal component extracted from the ΔOCS data, plotted at weekly
         intervals.  It shows a steady increase in OCS during winter and spring with a maximum around
         day 145, followed by a rapid decrease in summer with the maximum loss rate at day 200.  There

is little change in the autumn. The peak-to-peak amplitude is 5-6% of the total OCS column. This
         is more than double the 2.56±0.80 % peak-to-peak seen by Rinsland et al. [2002] from Kitt Peak,
         at a similar latitude to the majority of the MkIV data, and at an altitude within the range of the
         MkIV observations.  The inferred MkIV seasonal cycle is consistent with Wang et al.[2016] who
         reported xOCS data from five NDACC sites over the period 2005-2013.  As expected, the

amplitude of the MkIV seasonal cycle, which is representative of ~35ºN, is intermediate in value
         between that from the Jungfraujoch at 46°N (10-12% peak-to-peak) and Mauna Loa at 20°N (4-
         5%).

         Figure 6k shows the ΔOCS data with the SC subtracted.  This makes the LTC clearer.
         Fig 6l shows the ΔOCS data with the LTC subtracted, which greatly improves the compactness of

the seasonal behavior.  For example, the fact that the minimum OCS occurred in 2002, when the
         MkIV was taking measurements from 3.8 km altitude (red points) caused the red points to be
         systematically low in Fig. 6h, but this discrepancy disappears in Fig. 6l with the removal of the
         long-term changes.

         The greater fidelity of the data in Fig. 6l reveals some outliers. The blue data points,

measured from Fairbanks, Alaska during the summer of 1997, show a much larger drawdown of
         ΔOCS than is captured by the average seasonal cycle, which is representative of 35°N.  This may
         be related to the location of Fairbanks within the boreal forest.  Aside from this, there is
         remarkable consistency between the other sites, despite their huge range of altitudes.






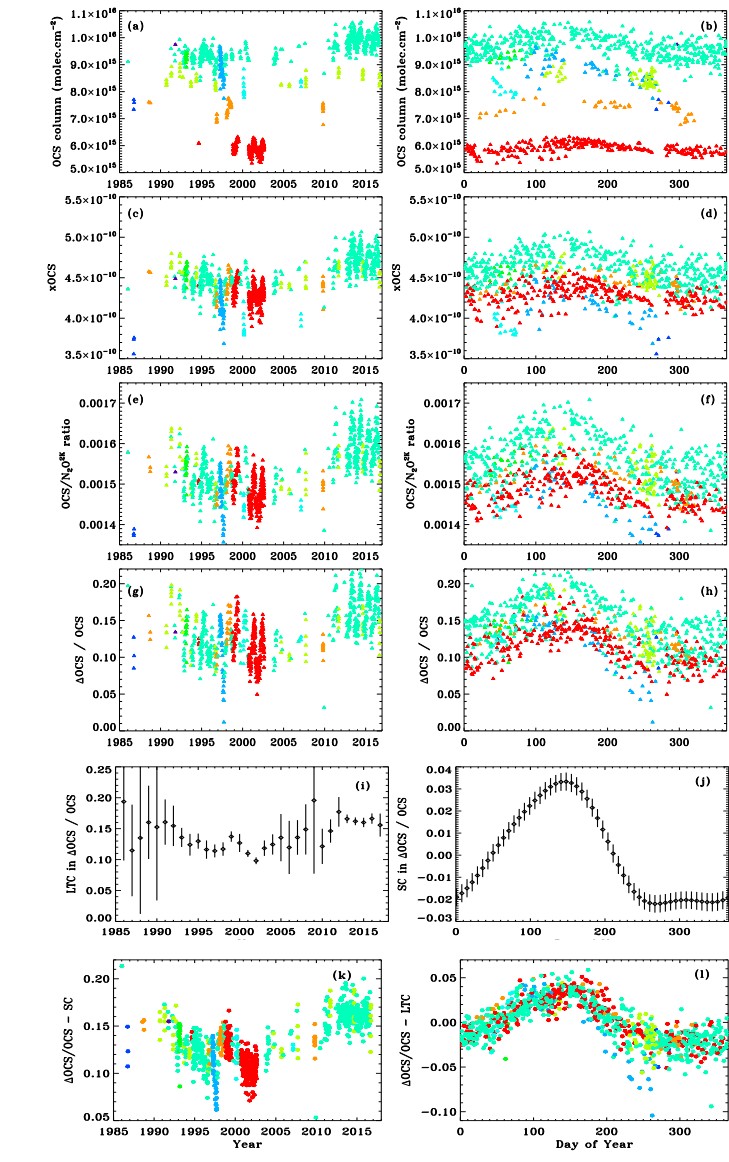

**Figure 6.** *Ground-based measurements of OCS, color-coded by site altitude (e.g., dark blue=0 km; light blue = 0.1 km; green=0.3 km; lime=1.2 km; orange =2.1 km; red=3.8 km). The left-hand panels show the OCS plotted versus year, revealing the long-term changes (LTC). The*

*right-hand panels show the same data plotted versus the day of the year, revealing the seasonal cycle (SC). Panels (a) and (b) show the raw column abundances, which are much larger at the lower altitude sites. Panels (c) and (d) show the column-averaged OCS amounts (xOCS), which reduces the altitude/site dependence. Panels (e) and (f) show the OCS/$N_2O$ column ratios, which further reduce the altitude/site dependence. Panels (g) and (h) show $\Delta$OCS, the difference*

*between the measured OCS, and the $N_2O$-based prediction described in Appendix A. Subtracting*



*the N$_2$O-derived OCS amount eliminates dynamically-induced variations common to both gases, revealing the tropospheric behavior with more fidelity. Panels (i) and (j) show the LTC and the SC extracted from the ΔOCS results by simultaneous fitting a linear spline and harmonic terms. Panel (k) shows the ΔOCS data with the SC subtracted. Panel (l) shows the SC with the LTC*
*subtracted.*

**Error Budget (ground-based)**

It is not straight-forward to categorize all errors in terms of random or systematic. While the contribution of measurement noise is clearly 100% random, and that of line intensities is 100% systematic, most errors sources have hybrid characteristics. That is, over a sufficiently
short timescale they can be considered invariant, but over longer time periods they become more random. For example, atmospheric temperature errors can be considered a fixed systematic error over a period of minutes, but their affect on measurements made hours and days apart is much more random. Table 4 attempts to quantify the uncertainties resulting from various error terms. In terms of the absolute accuracy of the measurements, all error terms contribute fully. In terms
of the precision, invariant errors (e.g., line intensities) do not contribute at all and the hybrid terms contribute partially.

Regarding the integrity of the seasonal cycle shown in figure 6j, the largest risk is site-to-site differences, coupled with the fact that measurements from certain sites (e.g. Ft. Sumner) only happen at particular times of year (September). For example, if the Ft Sumner xOCS were biased
low, then this would exaggerate the seasonal cycle because September happens to be the minimum of the seasonal cycle. But the fact that the de-trended data in 6l show good site-to-site consistency (apart from Fairbanks, Alaska) reduces the possibility of a large site-to-site bias.

| Error Type | Absolute Accuracy (%) | Precision (%) |
|---|---|---|
| Spectroscopy | | |
| OCS Intensities | 5 | 0 |
| OCS Air-broadening | 5 | 2 |
| Interfering gases | 4 | 2 |
| Interfering solar | 2 | 0 |
| ILS | 2 | 1 |
| Spectrum errors | | |
| Zero Level Offsets | 2 | 1 |
| Phase Errors | 2 | 2 |
| Channel Fringes | 2 | 1 |
| Ghosts | 1 | 1 |
| Forward Model | 1 | 0 |



| Smoothing Error | | |
|---|---|---|
| OCS | 5 | 2 |
| Interferers | 5 | 2 |
| T/P profile | 4 | 3 |
| Measurement Noise | 1 | 1 |
| **RSS Total** | **12** | **6** |

**Table 4.** *Error budget of absolute accuracy of retrieved OCS amounts,*
*and the achievable precision. The latter is smaller due to stationary*
*errors, that are the same or similar in every retrieval.*

Of course, although invariant systematic errors (e.g., spectroscopic line intensities) drive up the
absolute uncertainty, they do not degrade our ability to determine trends. Spectroscopic line
widths, on the other hand, can change the derived trend due to the altitudes of the various sites,
and hence the pressure broadening. After consideration of the use of $N_2O$ to reduce the effects of
smoothing error from 5% to 2%, we estimate that the precision of these MkIV OCS
measurements is 6%. With many years of data with this precision, OCS changes as small as 3%
can be detected in MkIV ground-based data.

**Discussion**

Various groups have reported changes (or lack thereof) in atmospheric OCS over the past
two decades. Griffith et al [1998] reported "seasonal cycles in the OCS total columns from both
Lauder and Wollongong with peak-peak (p-p) amplitudes of 6% and 18%, respectively, with both
cycles peaking in late summer (mid-February). An apparent cycle amplitude of about 5% is
expected as a result of tropopause height variations, and the remainder can be ascribed to seasonal
cycles in tropospheric mixing ratios. The secular trend in OCS was < 1%/year". This implies that
the 5% seasonal cycle seen at Lauder was mainly due to tropopause height variations and that the
actual variation in tropospheric OCS mole fraction was only 1% at Lauder and 13% at
Wollongong.

In our analysis, the OCS variation due to tropopause height variation has already been
implicitly removed from the ΔOCS by the $N_2O$ correction, since the tropopause height variations
also influence $N_2O$. So our 5-6% seasonal variation can be considered intermediate between the
Lauder (1%) and Wollongong (13%) values.

Rinsland et al. [2002] reported a trend of -0.25±0.04 %/year in the OCS column below 10
km above Kitt Peak over the period 1978 to 2002 with a seasonal cycle of 1.28±0.40% amplitude.
This was based on fitting a straight line to the long-term trend and a two-coefficient seasonal
cycle with a one-year period.



Kremser et al. [2015] report OCS increases of 0.5%/year from three SH sites (34S, 45S, 78S) over the period 2001-2014. Kremser used a linear spline with user-selected knot points to

represent the long-term behavior, after removing the seasonal variation. Lejeune et al. [2017] reported a 4%/year increase in the OCS partial column from 13.8km to 19.5 km over the period 1995-2015. For the tropospheric partial column (3.6 - 8.9 km), they reported a 6% drop from 1995 to 2002.5, a 7% increase from 2002.5 to 2008, and flat since. This latter behavior is highly consistent with the behavior seen in the MKIV ground-based dataset.


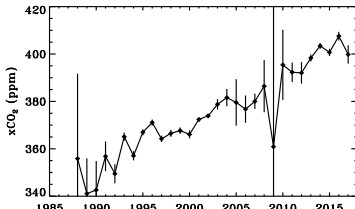 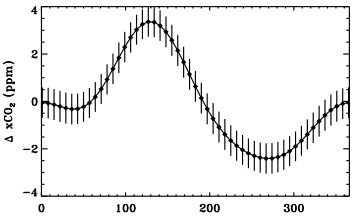

**Figure 7.** *Decomposition of MKIV ground-based measurements of $xCO_2$ into a long-term secular trend (left) and a seasonal cycle (right). Years with little data, or with inconsistent $xCO_2$ values, have large uncertainties. The seasonal cycle has its peak around day 130 and its maximum rate*

*of decrease around day 185.*

Figure 7 shows the long-term secular changes and seasonal cycle of xCO2, derived from the same MkIV observations. The $xCO_2$ underwent a similar analysis to that performed for OCS in figures 6i and 6j, with the exception that the $N_2O$ correction was not performed because it

would likely do more harm than good, given that the $CO_2$ profile decreases by only ~2% in the stratosphere (versus 100% for OCS). So the $N_2O$-related errors introduced would likely have been larger than that of the transport errors removed. The resulting $xCO_2$ seasonal cycle obtained is 5-6 ppm (1.4%) in peak-to-peak amplitude and has a similar shape to that of xOCS. This similarity is consistent with OCS being absorbed by plants during photo-synthesis. Upon closer

inspection of figure 7, the peak occurs around day 130 and the fastest $xCO_2$ loss occurs around day 185. These are each about two weeks earlier than that in xOCS.

The use of $N_2O$ to reduce the effects of stratospheric transport on OCS has been successfully applied. This relies on the fact that stratospheric transport affects both gases similarly. In terms of the ground-based measurements, $N_2O$ has the advantage over other tracers

(e.g. $CH_4$) that its tropospheric mole fraction is very stable with a seasonal variation of < 0.1%. Thus we can unambiguously attribute variation in $xN_2O$ at a particular site to the stratosphere.





Moreover, $N_2O$ can be measured to a high precision over a wide range of measurement conditions.

The MkIV balloon measurements show no significant trend at the $N_2O^{2K}$=250 ppt
isopleth (see fig 3c), which corresponds to ~21 km altitude at mid-latitudes, or at any other level. Over the 1995-2015 period, the change in MkIV OCS was -1±3%, as compared with 4±1% from Lejeune. These estimates do not quite overlap, but given that the altitudes and latitudes are different, the small discrepancy is not a cause for concern.

Historically, atmospheric trace gas abundances were retrieved using narrow windows
centered on isolated absorption lines of the gas of interest. This was computationally fast and avoided the worst interferences. From the $v_3$ OCS band, just 2 or 3 of the cleanest OCS lines were typically utilized. Faster computers now make it possible to fit far wider windows containing many more OCS lines, promising improved precision. The question is: does this improve the OCS retrieval? The answer depends on the quality of the radiative transfer
calculation, including the atmospheric T/P/Z and VMR profiles, and the spectroscopy, in particular that of the interfering lines. For example, including an OCS line overlapped by a T-sensitive $H_2O$ line will make the retrieval sensitive to lower tropospheric temperature errors. But if the temperature model is accurate, then adding the overlapped OCS line will nevertheless improve the retrieval.

Besides improving precision due to utilization of more target lines, wide windows have other benefits. Retrievals are more robust than from narrow windows in the sense that you are less likely to get a good fit (and hence a small uncertainty) for the wrong reasons. And the likelihood of non-convergence is reduced. Regions blacked out by $CO_2$ and $H_2O$ lines, although containing no information about the target gas, allow correction of zero offsets, which affect
stronger interfering gases, which in turn affect the target gas retrievals. So while the direct effect of zero offset on a weakly absorbing target gas is small, the indirect effect can be much larger. Broad windows also facilitate the identification and correction of channel fringes, although this was not necessary in the OCS windows in the MkIV spectra. Broad windows also allow a more accurate estimate of the doppler shift of the solar Fraunhofer lines, of which there are many
(arising from solar CO) in the 2000-2100 $cm^{-1}$ region. These doppler shifts cannot be accurately calculated since a large component arises from mis-pointing of the solar tracker.

We are not claiming that broad windows are always better than multiple narrow windows. It depends on how well the overlapping interferences can be accounted for. So this must be decided on a case-by-case basis and will depend on the quality of the spectroscopy and



the a priori T/P/VMR profiles. The altitude of the observation site(s) can also be important,
especially for windows containing $H_2O$ absorptions.

The narrow window approach avoids potential dangers (interferences) without
determining whether they are damaging or benign. Our approach is based on the results
themselves: the size of the uncertainties computed from the spectral fits, the actual precision of

the measurements, and the correlations between windows. While none of these metrics is
individually perfect, they collectively provide a rigorous screening.

**Summary & Conclusions**

We have retrieved OCS from over 30 years of MkIV balloon and ground-based spectra.
Simultaneous measurements of $N_2O$ were used to reduce the effects of stratospheric transport on

the OCS amounts, yielding better information on the tropospheric trends and seasonal cycle. This
makes no assumptions about the tropopause altitude or the stratospheric profile of OCS (other
than its relationship with $N_2O$). Balloon results yield no significant stratospheric trend.
Tropospheric OCS, on the other hand, shows a 5% decrease during 1990-2002, followed by a 5%
increase from 2003-2012. There was no discernible change since 2012. The reasons for this

behavior are not fully understood. Lejeune et al. [2017] speculate that this is partly due to a
reduction in industrial OCS-forming emissions in the 1990s.

We have also derived a tropospheric seasonal cycle which is 5-6% of the total column at
35N and much larger at 65N, the latter base on 1997 measurements from Fairbanks, Alaska. The
OCS seasonal cycle is similar in shape to that of $CO_2$, implying uptake by plants during

photosynthesis, but is 4-5 times larger, expressed as a fraction of the total column.

The fact that the balloon and ground-based measurements were taken with the same
instrument and analyzed with the same software (phase correction, FFT, spectral fitting) and
linelists provides the best possible internal consistency of results.

This work also highlights the advantages of using wide windows containing 20-30 lines

of the target gas, versus the traditional NDACC strategy of using 2-3 narrow windows, each
containing a single well-isolated target line. Despite the rms spectral fits being typically a factor
2 worse for the wide windows (due to interfering absorptions), the precisions of the retrieved
target gases are generally superior than for narrow, single-line windows. This is mainly because
the increased number of target lines out-weighs the degradation of the spectral fits. Also, the

presence of saturated interfering lines allows an accurate retrieval of any zero level offset. The
wide windows allow a better characterization of any channel fringes in the spectra and solar
doppler shifts.



In the future, as we improve our ability to model atmospheric radiative transfer (i.e. spectroscopic parameters, atmospheric P/T analyses), we can anticipate the systematic residuals

decreasing, allowing the broad windows to perform even better.  In contrast, the narrow windows, already being fitted close to the spectral noise level, won't improve as much.

**Highlights:**

- We report 29-year time series of atmospheric OCS measurement made from balloon and the ground by the same instrument.  Since most Stratospheric Sulphate Aerosol (SSA) is believed to come from OCS, accurate measurements of OCS are important to understand the behavior of SSA.
- We use measured N2O columns, together with a stratospheric OCS/N2O relationship established from balloon measurements, to correct the OCS columns for transport-driven variations of stratospheric OCS, to more clearly see the variations in tropospheric OCS.

- No variations seen in stratospheric OCS, despite a 5% minimum in tropospheric OCS around 2002/3

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



## Appendix A:  Correcting for Stratospheric Transport by use of $N_2O$

The OCS/$N_2O$ relationship derived from MkIV balloon profiles (Figure A.1) is linear for
$N_2O > 100$ ppb.  A "knee" occurs at $N_2O =120$ ppt which corresponds to the $P_b=10$ mbar pressure
level at mid-latitudes or an altitude of ~30 km.  Assume that

$$OCS(p) = a[N_2O^{2K}(p)\text{-}b] \qquad \text{for } N_2O^{2K} > b \qquad\qquad (A.1a)$$

$$OCS(p) = \quad 0 \qquad\qquad \text{for } N_2O^{2K} < b \qquad\qquad (A.1b)$$

p is the pressure, b=120 ppb, a=0.00225 is the gradient of the linear part of the OCS-$N_2O^{2K}$ curve.
$N_2O^{2K}(p) = N_2O(p)/(1+0.0025(\text{year}-2000))$ where 0.0025 is the rate of increase of $N_2O$.

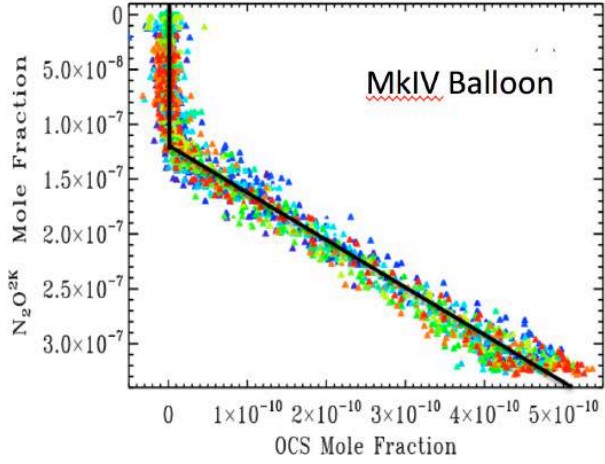

*Figure A.1.* *The MKIV balloon OCS-$N_2O$ relationship color-coded by year and the fitted straight-line functions (black).*

The $N_2O$ column is the integral of the $N_2O$ mole fraction with respect to pressure

$$C_{N2O}^{2K} = \int_0^{Ps} N_2O^{2K}(p)\,dp \qquad\qquad (A.2)$$

The OCS column is the integral of the OCS mole fraction with respect to pressure

$$C_{OCS} = \int_0^{Ps} OCS(p)\,dp \qquad\qquad (A.3)$$

where $P_s$ is the surface pressure in units of molec.cm$^{-2}$.

Substituting for OCS(p) from equations (A.1) yields

$$C_{OCS} = \int_{Pb}^{Ps} a[N_2O^{2K}(p) - b]\,dp \qquad\qquad (A.4)$$

where the integration limits are now $P_b$ to $P_s$ since OCS is zero between 0 and $P_b$.

Assuming a linear relationship between $N_2O$ and pressure between p=0 and $P_b$, as seen in figure
A.2, then the $N_2O$ column above $P_b$ is $P_b.b/2$, which means that the OCS column is

$$C_{OCS} = a[C_{N2O}^{2K} - b(P_s - P_b/2)] \qquad\qquad (A.5)$$




Since $P_b$ is only ~10 mbar as compared with ~1000 mbar for $P_s$, the second term *[b.(P_s-P_b/2)]* in A.5 is only about a third of the size of the first term ($C_{N2O}$) and so the fractional change in $C_{OCS}$ for a doubling of $P_b$ is only 0.75%. So the $P_b$ term tends to be unimportant. This is another way of saying that the OCS-N$_2$O relationship is highly linear up to ~30 km altitude or 10 mbar, which

represents over 99% of the N$_2$O column and over 99.9% of the OCS column. So the non-linearity above 30 km, or uncertainty in the "knee" altitude, doesn't have an important impact on $C_{OCS}$.

By substituting the measured N$_2$O column, de-trended to the year 2000, into (A.5) we can predict the OCS column. This prediction encapsulates the stratospheric transport effects. By subtracting the predicted OCS from the actual measured OCS column

$$\Delta_{OCS} = C_{OCS}^M - C_{OCS}^p$$
$$\Delta_{OCS} = C_{OCS}^M - a[C_{N2O}^{2K} - b(P_s - P_b/2)] \tag{A.6}$$

we can derive an OCS anomaly, $\Delta_{OCS}$, representing variations in OCS occurring below the altitudes covered by the balloon-derived OCS-N$_2$O relationship. By dividing $\Delta_{OCS}$ by $C_{OCS}^M$, we get a dimensionless quantity. Positive values indicate that the tropospheric OCS is in excess of

the N$_2$O-based prediction.

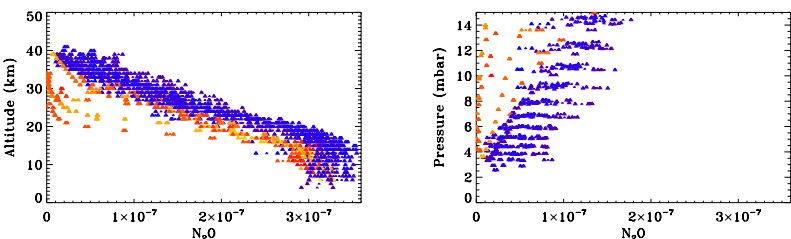

***Figure A.2.*** *MkIV N$_2$O profiles, color-coded by latitude. These look similar to the OCS profiles in figure 2b. The right panel shows the linear relationship between N$_2$O and pressure at the highest altitudes. The horizontal banding is an artifact of the 1 km vertical grid used in the*
*retrieval.*

### Appendix B: Selection of ground-based OCS windows

Initially, 21 OCS windows were defined and evaluated in each ground-based spectrum, all located in the strong $\nu_3$ band (See Table B.1). None of the weaker OCS bands (at 868, 2915 and 4096 cm$^{-1}$) were used in the ground-based analyses because their OCS absorptions are simply
too weak and/or overlapped with interfering absorptions. Of these 21 windows, 4 were new and 17 had been used previously. In the latter category are the two windows used by Griffith et al. [1998] for analysis of spectra from Wollongong and Lauder. Three windows used by Rinsland et al. [2002] for analysis of ground-based Kitt Peak spectra (2.1 km altitude), and subsequently used



by Mahieu et al. [2003] for analysis of JFJ spectra. Four windows used by Krysztofiak et al.

[2015] for analysis of ground-based OCS from Paris. Four windows used by Kremser et al.
[2015] for analysis of three SH sites. Four windows used by Lejeune et al., [2017] for analysis of
spectra from the Jungfraujoch (JFJ).

Table B.1 summarizes the attributes of the 21 tested windows, including their center
wavenumber, width, and fitted gases. Also included are the OCS line strengths (mean, max &

sum) and their mean Ground State Energy (E"). Figure B.1 plots the wavenumber of extents of
each window above a ground-based spectral fit to most of the OCS band. In all the 17
previously-used windows, there were several instances of close similarity: for example, the $P_{25}$
line at ~2051.32 cm$^{-1}$ was used by everyone except Griffith et al., [1998]. Surprisingly, the OCS
R-branch has never previously been used for ground-based OCS retrievals, to the best of our

knowledge, despite the lines being slightly stronger than those in the P-branch and more closely
spaced.

Lejeune et al. [2017] gave a detailed description of their window selection and
optimization process. Their initial selection was based on minimizing overlap with interfering
absorptions, especially $H_2O$, under the conditions experienced at the Jungfraujoch (JFJ, 3.58 km

altitude). Surprisingly, the OCS lines ranked 1'st (2055.86 cm$^{-1}$) and 4'th (2052.72 cm$^{-1}$) in
Lejeune's listing of telluric non-interference were not utilized in their analysis of the full 20 year
JFJ data-set. And the OCS line ranked 15'th (2054.53 cm$^{-1}$) *was* used by Lejeune et al., despite
being substantially overlapped by $H_2O$, which severely degrades this particular window at sites
that are significantly warmer or lower in altitude than JFJ. Their final window selection was

based on DOFs and information content.

In addition to these 17 previously-used windows, in the present work four new, much-
broader, windows were also evaluated. In these broad windows, most of the OCS lines are
overlapped by stronger interfering absorbers, which would disqualify them in the traditional
NDACC-IRWG window selection process. We argue here, however, that mere overlap with an

interfering absorption is not sufficient grounds for exclusion. Provided the residual is reasonably
small, an overlapped OCS line can still provide useful information.

Of the four new windows, the one centered at 2060 cm$^{-1}$ covers the whole band. This
includes some saturated $H_2O$ lines at 2060.48 and 2065.50 cm$^{-1}$ (see fig B.1) which tend to give
rise to large residuals affecting nearby OCS lines. When these saturated $H_2O$ lines are excluded

from the fits, by splitting the wide window into two, resulting in the broad windows centered at
2051 and 2070 cm$^{-1}$, the residuals improve considerably (from 0.55% to 0.40%) with only a small



loss of OCS information. This results in a better overall retrieval accuracy, and hence smaller uncertainties in the derived window-to-window biases.

| # (j) | Center (cm$^{-1}$) | Width (cm$^{-1}$) | Lines | Fitted Interfering Gases | $S_{max}$ x10$^{-18}$ | $\sum S$ x10$^{-18}$ | $S_{bar}$ x10$^{-18}$ | E" cm$^{-1}$ |
|---|---|---|---|---|---|---|---|---|
| 1 $^G$ | 2045.485 | 0.65 | P$_{37}$ | ocs o3 co2 co | 0.816 | 1.134 | 0.606 | 366 |
| 2 $^G$ | 2055.805 | 0.33 | P$_{15}$ | ocs o3 co2 co | 1.050 | 1.137 | 0.972 | 74 |
| 3 $^R$ | 2045.485 | 0.37 | P$_{37}$ | ocs o3 co2 co | 0.816 | 1.046 | 0.654 | 368 |
| 4 $^R$ | 2051.33 | 0.30 | P$_{25}$ | ocs o3 h2o co2 | 1.165 | 1.226 | 1.109 | 137 |
| 5 $^R$ | 2055.80 | 0.32 | P$_{15}$ | ocs o3 co2 co | 1.050 | 1.137 | 0.972 | 74 |
| 6 $^K$ | 2038.95 | 0.30 | P$_{50}$ | ocs o3 h2o co2 | 0.356 | 0.392 | 0.324 | 558 |
| 7 $^K$ | 2048.25 | 0.90 | P$_{32}$-P$_{31}$ | ocs o3 h2o co2 co | 1.030 | 2.530 | 0.826 | 270 |
| 8 $^K$ | 2051.40 | 0.40 | P$_{25}$ | ocs o3 h2o co2 co | 1.176 | 1.367 | 1.001 | 181 |
| 9 $^K$ | 2054.95 | 4.90 | P$_{22}$-P$_{12}$ | ocs o3 h2o co2 co | 1.183 | 13.61 | 0.966 | 117 |
| 10 $^S$ | 2048.00 | 0.44 | P$_{32}$ | ocs o3 co2 | 0.998 | 1.251 | 0.815 | 278 |
| 11 $^S$ | 2049.935 | 0.37 | P$_{28}$ | ocs o3 co2(2) co | 1.112 | 1.323 | 0.926 | 224 |
| 12 $^S$ | 2051.33 | 0.29 | P$_{25}$ | ocs o3 h2o co2 | 1.165 | 1.226 | 1.109 | 137 |
| 13 $^S$ | 2054.11 | 0.26 | P$_{19}$ | ocs o3 co2 h2o co | 1.157 | 1.222 | 1.096 | 109 |
| 14 $^L$ | 2048.045 | 0.39 | P$_{32}$ | ocs o3 co2 | 0.998 | 1.244 | 0.815 | 278 |
| 15 $^L$ | 2049.975 | 0.41 | P$_{28}$ | ocs o3 co2(2) co | 1.112 | 1.350 | 0.926 | 224 |
| 16 $^L$ | 2051.32 | 0.28 | P$_{25}$ | ocs o3 h2o co2 | 1.165 | 1.226 | 1.109 | 137 |
| 17 $^L$ | 2054.50 | 0.34 | P$_{18}$ | ocs o3 co2 h2o(2) | 1.139 | 1.234 | 1.054 | 101 |
| 18 $^T$ | 2051.30 | 13.10 | P$_{38}$-P$_{11}$ | ocs o3 h2o co2 co | 1.183 | 35.23 | 0.882 | 203 |
| 19 $^T$ | 2053.55 | 4.80 | P$_{25}$-P$_{15}$ | ocs o3 h2o co2 co | 1.183 | 14.26 | 1.018 | 134 |
| 20 $^T$ | 2060.17 | 30.95 | P$_{38}$-R$_{36}$ | ocs o3 h2o co2 co | 1.248 | 85.02 | 0.833 | 208 |
| 21 $^T$ | 2071.10 | 9.00 | R$_{11}$-R$_{36}$ | ocs o3 h2o co2 co | 1.248 | 33.96 | 0.978 | 204 |

***Table B.1.*** *Attributes of the 21 ground-based OCS spectral windows evaluated on MkIV ground-based spectra using the GFIT algorithm. Center and Width are in units of cm$^{-1}$. Windows 1-17 represent the old, previously-used, windows and are grouped chronologically by reported use. Windows 1-2 were defined by Griffith et al,[1998]; 3-5 were defined by Rinsland et al.[2002]; 6-9 by Krysztofiak et al. [2015]; 10-13 by Kremser et al. [2015]; and 14-17 by Lejeune et*

*al.[2017]. Windows 18-21 are the new ones. The fitted OCS lines are all from the $v_3$ band and the column labeled "Lines" shows the spectroscopic assignment of the strongest OCS lines in each window (exceeding 1% absorption depth). The number of such lines is simply the difference of the quantum numbers plus one. So window #20 contains 38+36+1 = 75 lines. Window #21 contains 36-11+1=26 lines. $S_{max}$ is the maximum OCS line intensity in units of cm$^{-1}$/(molec.cm$^{-2}$),*

*$\sum S$ is the sum of intensities, and $S_{bar}$ is the mean (S-weighted) intensity. E" is the mean (S-weighted) ground state energy. The attributes tabulated above are all independent of the measured spectra.*

All 21 candidate OCS windows were run through the full ground-based MkIV dataset,

comprising over 1000 observation days at 12 different sites. A statistical analysis was then performed on the column amounts retrieved by the GFIT algorithm. Each window was assumed to have a scale factor, $\Lambda_j$, such as might arise due to multiplicative errors in the spectral line





strengths. And for each spectrum there is an average value of the retrieved geophysical quantity $\overline{Y}_i$, which might represent a vertical column abundance or the scale factor that multiplies the a

priori VMR profile. $\Lambda_j$ and $\overline{Y}_i$ are found by iteratively minimizing the quantity

$$\chi^2 = \sum_{j=1}^{NW} \sum_{i=1}^{NS} \left( \frac{Y_{i,j} - \Lambda_j \overline{Y}_i}{\varepsilon_{i,j}} \right)^2$$

where i is an index over spectra and j is an index over windows. NS is the number of spectra. NW is the number of windows, 21 in the case of OCS. $Y_{i,j}$ is the measured value (e.g. column abundance) retrieved from the i'th spectrum using the j'th window of a particular gas, and $\varepsilon_{i,j}$ is

its uncertainty (basically the square root of the diagonal element of $(K^T S_y K)^{-1}$, where K is the Jacobian matrix and $S_y$ is the measurement covariance, estimated from the spectral fits). Hence the minimization obtains NW+NS unknowns from NW*NS data points, so the solution is fully determined provided that NW≥2 and NS≥2. Since $\Lambda_j$ and $\overline{Y}_i$ multiply each other, the problem is non-linear and so the solution must be found iteratively. If there are no biases between windows,

$\Lambda_j$ will be equal to the unit vector, and so the equation reduces to the usual definition of the mean: the value that minimizes the standard deviation of the points about it. If the uncertainties $\varepsilon_{i,j}$ are a true representation of the scatter of $Y_{i,j}$ about the mean, then the term in parentheses will have an average value of ~1, and so the $\chi^2$ will have a value of NW*NS=N$_T$.

The results of this analysis are presented in Table B2. Firstly, the average rms spectral

residuals (Fit %) are reported for each window. These tend to be small for the narrow windows and larger for the wide windows. This is because the narrow windows were previously optimized to avoid large residuals. The average scale factor ($\Lambda_j$) of each window was computed with respect to the mean of all 21 windows. Departures from the ideal value of 1.0 can be considered a bias. The $\overline{\varepsilon}_j$ tells us the average uncertainty in the retrieved OCS column from the j'th window.






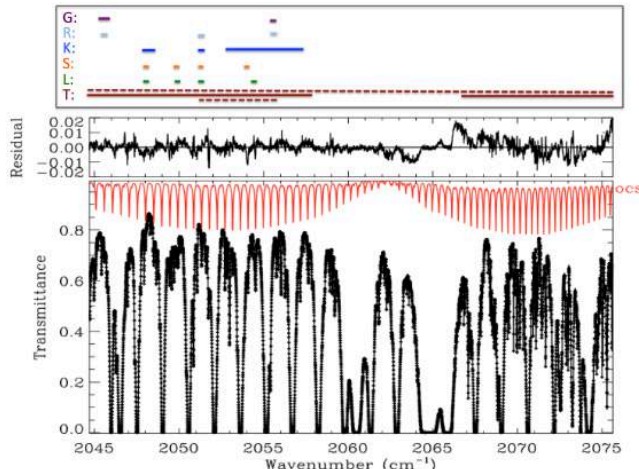

***Figure B.1.*** *Spectral coverage of the evaluated windows superimposed on a fit to a ground-based MkIV spectrum measured from JPL at 58° SZA. The black points and line are the measured and calculated spectra, respectively. The red trace shows the OCS absorption, the strong lines extend from $P_{38}$ on the left to $R_{36}$ on the right. In the top panel, the horizontal bars show the coverage of the windows evaluated in the study. "G" represents the windows used by Griffith et al. [1998], "R" Rinsland et al. [2002], and so on. The brown solid lines labeled "T" were considered the best. The brown dotted lines were evaluated but rejected. This figure omits window #6 centered on the $P_{50}$ line at 2038.95 $cm^{-1}$, used by Kryszofiak et al.[2015].*

We also computed the $\chi^2/N_T$, the factor by which the scatter of the measurements compares with their uncertainties estimated from the spectral fits ($\bar{\varepsilon}_j$). In a perfect world $\chi^2/N_T$ would be close to 1.0. Values of $\chi^2/N_T$ that are significantly less than 1 imply a persistent systematic error in the spectral fits (e.g. line position error) that drives up $\bar{\varepsilon}_j$, but does not impair the precision of the measurements. Values of $\chi^2/N_T$ that are significantly >1 imply a hidden factor that increases the spectrum-to-spectrum scatter of the OCS measurements without impacting the quality of the spectral fits. This could be an interfering T-dependent $H_2O$ absorption blended with the OCS line. So $\bar{\varepsilon}_j \chi^2/N_T$ provides an estimate of the precision of the measurements.

We also computed correlation coefficients (CC) between the 21 evaluated ground-based OCS windows and each other, and with the mean. In the interest of Table B.2 being printable, we omit the 21 columns containing the window-to-window CCs and show only the window-to-mean CC. Note that these are the correlations in the variations of xOCS. Total column OCS would produce much larger CCs due to the fact that surface pressure changes (primarily driven by



changes in observation altitude) would induce additional highly-correlated changes between

windows.

| # (j) | Center (cm$^{-1}$) | Width (cm$^{-1}$) | Fit (%) | $\Lambda_j$ | $\bar{\varepsilon}_j$ | $\chi^2/N_T$ | $\bar{\varepsilon}_j\chi^2/N_T$ | CC |
|---|---|---|---|---|---|---|---|---|
| 1$^G$ | 2045 | 0.65 | 0.190 | 0.964 | 0.030 | 0.73 | 0.022 | 0.790 |
| 2$^G$ | 2055 | 0.33 | 0.215 | 1.013 | 0.029 | 0.57 | 0.017 | 0.827 |
| 3$^R$ | 2045 | 0.37 | 0.171 | 0.952 | 0.036 | 0.66 | 0.024 | 0.811 |
| 4$^R$ | 2051 | 0.30 | 0.148 | 0.997 | 0.014 | 0.76 | 0.011 | 0.798 |
| 5$^R$ | 2055 | 0.32 | 0.215 | 1.015 | 0.030 | 0.57 | 0.017 | 0.827 |
| 6$^K$ | 2038 | 0.30 | 0.125 | 0.909 | 0.053 | 0.75 | 0.040 | 0.702 |
| 7$^K$ | 2048 | 0.90 | 0.229 | 0.991 | 0.015 | 1.15 | 0.017 | 0.786 |
| 8$^K$ | 2051a | 0.40 | 0.181 | 0.995 | 0.017 | 0.92 | 0.016 | 0.774 |
| 9$^K$ | 2054 | 4.90 | 0.373 | 1.014 | 0.017 | 0.70 | 0.011 | 0.829 |
| 10$^S$ | 2048a | 0.44 | 0.196 | 0.984 | 0.020 | 0.67 | 0.013 | 0.835 |
| 11$^S$ | 2049 | 0.37 | 0.195 | 1.011 | 0.025 | 0.63 | 0.016 | 0.827 |
| 12$^S$ | 2051b | 0.29 | 0.144 | 0.997 | 0.014 | 0.74 | 0.011 | 0.794 |
| 13$^S$ | 2054a | 0.26 | 0.150 | 0.986 | 0.030 | 0.77 | 0.023 | 0.791 |
| 14$^L$ | 2048b | 0.39 | 0.182 | 1.002 | 0.023 | 0.78 | 0.018 | 0.825 |
| 15$^L$ | 2049a | 0.41 | 0.190 | 1.012 | 0.025 | 0.62 | 0.015 | 0.835 |
| 16$^L$ | 2051c | 0.28 | 0.145 | 0.998 | 0.014 | 0.65 | 0.010 | 0.803 |
| 17$^L$ | 2054b | 0.34 | 0.195 | 0.981 | 0.051 | 1.09 | 0.056 | 0.640 |
| **18$^T$** | **2051d** | **13.1** | **0.394** | **1.009** | **0.015** | **0.54** | **0.008** | **0.840** |
| 19$^T$ | 2053 | 4.80 | 0.395 | 1.024 | 0.018 | 0.65 | 0.012 | 0.846 |
| 20$^T$ | 2060 | 30.9 | 0.552 | 0.987 | 0.021 | 0.49 | 0.010 | 0.855 |
| **21$^T$** | **2071** | **9.0** | **0.417** | **1.005** | **0.021** | **0.33** | **0.007** | **0.831** |

***Table B.2.*** *Statistical properties of the OCS retrievals from the 21 investigated windows. Center
and Width are as in Table B.1. Fit is the average rms spectral fitting residual achieved in that*
*particular window. Of course, this depends on the spectra that are fitted (their altitude, SZA,
spectral resolution, etc.) and so its absolute value is somewhat arbitrary, but its window-to-
window variation is significant since all windows were fitted in all spectra. $\Lambda_j$ is the average
value of the OCS VMR Scale Factor retrieved from window j, relative to the mean of all windows.
$\bar{\varepsilon}_j$ is an estimate of the mean uncertainty associated with retrieving OCS from window j, based on*
*the spectral fits. $X^2/N_T$ represents the average value of the ratio of the scatter of the
measurements from the mean, to the estimated uncertainty. $\bar{\varepsilon}_j\chi^2/N_T$ represents the precision of
the measurements. CC contains correlation coefficients of xOCS variations between each
evaluated ground-based window and the mean of all windows. Based on all the above, OCS
columns retrieved from windows #18 and 21 were selected and averaged for subsequent analysis,*
*with the other 19 windows discarded.*

        We now discuss how the information in Table B.2 is used to select the best windows.

Regarding the $\Lambda_j$ values, windows #1, 3, and 6 are clearly biased low by more than their average

uncertainties ($\bar{\varepsilon}_j$). And window #19 is biased high. These significant biases imply a problem with

the spectroscopy, and so it would be a risk to use these windows until the cause of the bias is





identified. So these 4 windows are rejected. In terms of their $\bar{\varepsilon}_j$ values, windows #6 and #17 both exceed 5%, and although there is nothing inherently malign in such high values, these windows will eventually be out-weighed by the others with much smaller uncertainties during the subsequent averaging over windows. So since these windows will negligibly impact the final

OCS results, they are better omitted. The $\chi^2/N_T$ values exceed 1 for windows #7 and #17, implying that the spectral fits under-estimate the true precision of the window, which is a cause for concern.

We should point out that all the parameters in Table B.2 except $\bar{\varepsilon}_j$ depend on comparing a particular window with the mean of all windows. So when the list of windows includes some

very similar variants on the same window (e.g., 2015 cm$^{-1}$), this has a disproportionate weight on the average and tends to increase the CC and brings the $\Lambda_j$ closer to 1.0. And since 19 of the windows occupy the P-branch and only one solely utilizes the R-branch, we should expect this window to have a poorer bias and a lower CC. So it is a pleasant surprise to see the window #21 do so well, with a bias of <1% and a CC of 0.83. And its $\chi^2/N_T$ value of 0.33 is the lowest of all

windows, implying that although systematic errors increased the residuals and hence $\bar{\varepsilon}_j$, this did not affect the precision, which at 0.007% is the best of all windows.

Of the four Lejeune windows, window #17 containing the $P_{18}$ line at 2054.50 cm$^{-1}$ stands out as clearly the worst, at least for the purpose of analyzing MkIV spectra. This is based on multiple factors including: a 1.9% bias in its $\Lambda_j$, its large (5%) uncertainty, its $\chi^2/N_T$ exceeding 1,

and a CC of only 0.64, which is the worst in the entire table. This poor performance is likely related to the contamination of this window by $H_2O$ in the lower altitude MkIV spectra. This window was rated as fifteenth best in Table 1 of Lejeune et al., [2017], an assessment with which we concur, but was nevertheless one of four eventually selected for use.

Window #6 containing the $P_{50}$ line at 2038.95 cm$^{-1}$ is another bad one. There is a 9.1%

low bias in the retrieved OCS amounts, the worst of all windows, and the uncertainties are over 5%, on average, despite having the smallest rms residual. Its CC of 0.70 is the second worst. We suggest that the poor performance of this window relates to the weakness of the $P_{50}$ OCS line, and it large E" value (558 cm$^{-1}$).

The $P_{15}$ line at 2055.8 cm$^{-1}$ was used by Griffith et al. (window #2) and Rinsland et al.

(window #5), but not by Kremser et al. nor Lejeune et al., despite this window being rated as first of the 21 in Table 1 of Lejeune et al. In our work we find this to be one of the better narrow windows, but not the best.

The best narrow windows were those centered on the $P_{25}$ line at 2051.4 cm$^{-1}$. Considering that it contains only one OCS line, the various narrow 2051 windows do remarkably





well. Its spectral fits are more than a factor 2 better than the wide 2051.30 cm$^{-1}$ window

containing 28 lines. Consequently the computed $\bar{\varepsilon}_j$ of the narrow and wide 2051 cm$^{-1}$ windows is

the same. The wide window has lower $\chi^2/N_T$ and consequently higher precision, and superior

correlation coefficients.

     Based on Table B.2, we opted to use the two broad windows centered at 2051 and 2071

cm$^{-1}$ (#18 and #21, bold and italicized) for the final OCS retrievals whose results are used in the

paper. Of the four wide windows in the lower half of the table, these two give the $\Lambda_j$ values

closest to 1.0 and have the best precisions ($\bar{\varepsilon}_j \chi^2/N_T$).

**Appendix C: Selection of ground-based N$_2$O windows**

     A study was performed on fifteen candidate N$_2$O windows covering 2400 to 4800 cm$^{-1}$.

These windows include: (1) three traditional, narrow, MIR windows used by NDACC, (2) nine

recently-defined broader MIR windows, and (3) three broad SWIR windows used by TCCON.

The table below defines these 15 windows and the fitted parameters, together with their key

attributes.

| # | Center (cm$^{-1}$) | Width (cm$^{-1}$) | N$_2$O Band | Lines | Fitted Interfering Gases | $S_{max}$ | $\sum S_i$ | $S_{bar}$ | E" cm$^{-1}$ |
|---|---|---|---|---|---|---|---|---|---|
| 1 | 2443.10 | 2.60 | $\nu_3+2\nu_2$ | $P_{23}$-$P_{21}$ | co2 | 4.56 | 15.9 | 3.67 | 312 |
| 2 | 2481.85 | 1.30 | $\nu_3+2\nu_2$ | $R_{23}$ | | 4.40 | 9.7 | 3.83 | 322 |
| 3 | 2806.32 | 0.44 | $\nu_1+\nu_2$ | $R_9$ | ch4 | 0.77 | 0.8 | 0.77 | 38 |
| 4 | 2446.00 | 26.20 | $\nu_3+2\nu_2$ | $P_{33}$-$P_4$ | co2 ch4 hdo h2o | 5.08 | 135. | 3.52 | 257 |
| 5 | 2479.70 | 19.80 | $\nu_3+2\nu_2$ | $R_9$-$R_{33}$ | co2 ch4 hdo h2o | 5.48 | 123. | 3.99 | 278 |
| 6 | 2539.80 | 46.60 | $2\nu_3$ | $P_{47}$-$P_1$ | hdo h2o ch4 co2 | 22.5 | 619. | 15.4 | 252 |
| 7 | 2580.40 | 34.60 | $2\nu_3$ | $R_0$-$R_{51}$ | hdo h2o ch4 co2 | 24.3 | 713. | 15.9 | 261 |
| 8 | 2781.70 | 25.20 | $\nu_1+\nu_2$ | $P_{31}$-$P_5$ | hdo h2o ch4 co2 o3 hcl | 0.71 | 0.3 | 0.35 | 436 |
| 9 | 2796.95 | 5.30 | $\nu_1+\nu_2$ | $Q_{39}$-$R_0$ | hdo h2o ch4 co2 o3 hcl | 1.59 | 41.0 | 1.16 | 191 |
| 10 | 2813.00 | 26.80 | $\nu_1+\nu_2$ | $R_1$-$R_{38}$ | hdo h2o ch4 co2 o3 hcl | 0.88 | 24.2 | 0.62 | 230 |
| 11 | 3344.40 | 2.48 | $\nu_1+2\nu_2$ | $P_{23}$-$P_{21}$ | 1h2o h2o co2 hcn | 1.36 | 4.2 | 1.22 | 251 |
| 12 | 3372.70 | 2.20 | $\nu_1+2\nu_2$ | $R_9$-$R_{11}$ | 1h2o h2o co2 hdo | 1.53 | 4.6 | 1.40 | 101 |
| 13 | 4395.20 | 43.40 | $2\nu_1$ | $P_{39}$-$P_1$ | ch4 h2o hdo | 1.13 | 33.2 | 0.72 | 278 |
| 14 | 4430.10 | 23.10 | $2\nu_1$ | $R_1$-$R_{49}$ | ch4 h2o hdo co2 | 1.17 | 30.1 | 0.86 | 196 |
| 15 | 4719.45 | 73.00 | $2\nu_1+\nu_2$ | $P_{42}$-$R_{56}$ | ch4 h2o co2 | 0.73 | 40.0 | 0.47 | 261 |

*Table C.1. Attributes of the 15 ground-based N$_2$O spectral windows that were tested using MkIV*
*ground-based spectra. Center and Width are in units of cm$^{-1}$. Band is the N$_2$O band with the*
*dominant lines in this window. Lines is the transition quantum numbers. $S_{max}$ is the maximum*
*N$_2$O line intensity in units of $10^{-21}$ cm$^{-1}$/(molec.cm$^{-2}$), $\sum S$ is the sum of intensities, and $S_{bar}$ is the*
*mean (S-weighted) intensity. E"$_{bar}$ is the mean (S-weighted) ground state energy in units of cm$^{-1}$.*

*The above attributes are all independent of the measured spectra. Blue denotes windows that*
*were subsequently rejected.*



No windows from the super-strong $N_2O$ $v_1$ band centered at 2224 $cm^{-1}$ were evaluated. These are saturated in ground-based spectral even at low airmasses. The windows at 2539 and

2580 $cm^{-1}$ contain the strongest lines, a factor 4 stronger than the four next strongest windows at 2400-2500 $cm^{-1}$ region. The 2781 $cm^{-1}$ window contains the weakest lines, closely followed by the 4719 $cm^{-1}$ TCCON window. Our philosophy is to have windows with a range of different line strengths to cover low- and high-airmass conditions. But here there is another consideration: we want to $N_2O$ averaging kernels to match those of OCS, which requires favoring the windows with

the weaker $N_2O$ lines.

These 15 windows were run through the MkIV ground-based dataset (4000+ spectra and 1000+ observation days), covering 12 different sites from 0 to 3.8 km altitude over 1985-2016. The table below summarizes the results for each window. Note that all windows were measured simultaneously in the same InSb spectrum.

$\Lambda_j$ is the mean scale factor for that window, relative to the mean value for all windows. The $\bar{\varepsilon}_j$ values show the average retrieval uncertainty for window j. The $\chi^2/N$ values are the actual variations of the measured $Y_{i,j}$ values divided by their estimated uncertainties $\varepsilon_{i,j}$. In a perfect world these would be 1.0. Values smaller than one indicate that the estimated uncertainties (computed by GFIT) are too large. This usually indicates a systematic spectral fitting residual

that increases the rms fit, and hence the uncertainty, but does not cause variations in the retrieved column values.

| # | Center | Width | Fit % | $\Lambda_j$ | $\bar{\varepsilon}_j$ | $\chi^2/N_T$ | $\bar{\varepsilon}_j\chi^2/N_T$ | CC |
|---|--------|-------|-------|------|-------|---------|----------|-----|
| 1[*] | 2443 | 2.6 | 0.136 | 0.9914 | 0.0052 | 0.399 | 0.0021 | 0.758 |
| 2[*] | 2481 | 1.3 | 0.148 | 0.9938 | 0.0053 | 0.463 | 0.0025 | 0.760 |
| 3 | 2806 | 0.4 | 0.138 | 0.9907 | 0.0133 | 0.906 | 0.0120 | 0.591 |
| 4[*] | 2446 | 26.2 | 0.205 | 0.9925 | 0.0066 | 0.319 | 0.0019 | 0.802 |
| 5[*] | 2479 | 19.8 | 0.187 | 0.9949 | 0.0055 | 0.361 | 0.0020 | 0.800 |
| 6 | 2539 | 46.6 | 0.277 | 1.0053 | 0.0080 | 0.838 | 0.0067 | 0.770 |
| 7 | 2580 | 34.6 | 0.349 | 1.0070 | 0.0101 | 0.612 | 0.0062 | 0.785 |
| 8[*] | 2781 | 25.2 | 0.414 | 1.0038 | 0.0164 | 0.644 | 0.0100 | 0.758 |
| 9 | 2796 | 5.3 | 0.416 | 1.0553 | 0.0142 | 0.888 | 0.0130 | 0.743 |
| 10[*] | 2814 | 29.8 | 0.434 | 1.0068 | 0.0167 | 0.498 | 0.0084 | 0.789 |
| 11[*] | 3344 | 2.5 | 0.208 | 1.0241 | 0.0112 | 0.662 | 0.0073 | 0.604 |
| 12[*] | 3372 | 2.2 | 0.190 | 1.0329 | 0.0089 | 0.833 | 0.0074 | 0.620 |
| 13[*] | 4395 | 43.4 | 0.374 | 0.9874 | 0.0141 | 0.490 | 0.0069 | 0.767 |
| 14[*] | 4430 | 23.1 | 0.328 | 0.9895 | 0.0121 | 0.537 | 0.0063 | 0.755 |
| 15[*] | 4719 | 73.1 | 0.297 | 0.9990 | 0.0114 | 1.010 | 0.0 | 0.662 |

*Table C.2. Statistical properties of the $N_2O$ retrievals and $xN_2O$ amounts retrieved from the 15 investigated windows. In the # column, a * symbol denotes that results from this window were*

*used in the subsequent analyses. So eleven windows were found acceptable and four not.*



*Window-to-window biases of retrieved $N_2O$ are presented ($S_j$) and the mean uncertainty of each window ($\bar{\varepsilon}_j$). The CC values are generally smaller than those of OCS, reflecting the smaller variations in atmospheric $xN_2O$ in comparison with the measurement noise.*

The 2796 window (#9) has the most deviant $\Lambda_j$ value (1.055). This window covers the Q-branch of the $v_1+v_2$ band. The adjacent windows cover the P- and R-branches of the same band, but are bias-free. So perhaps the Q-branch is affected by line mixing. Certainly the individual Q-branch lines are well overlapped in ground-based observations, and balloon observations (lower pressure, less LM) reveal only a 1% bias.

Table C.2 also shows correlation coefficients between the retrieved $xN_2O$ from each of the fifteen $N_2O$ windows, and the mean. A high CC does not necessarily mean that a window is good. It just means that it consistently shows similar behavior to the mean of all windows.

The broad 2446 and 2479 cm$^{-1}$ windows (# 4,5) have the highest CCs with mean values $\geq$ 0.8. The 2806 cm$^{-1}$ window (#3) has the worst CC with a mean value of 0.59, which is probably
related to its low E"_bar (=38 cm$^{-1}$). Temperature-independent transitions have an E" of ~300 cm$^{-1}$, so an E" of 38 cm$^{-1}$ will allow errors in the assumed atmospheric temperature to induce additional variations in the retrievals, not present in the other less T-sensitive windows. This will drive down the CC, and increase the $\chi^2/N_T$.

It is perhaps surprising that the narrow windows containing a single $N_2O$ line do so well
in terms of their $\bar{\varepsilon}_j$. This is because you can achieve very good fits to a narrow window by avoiding large residuals due to poor spectroscopy or other factors (e.g., temperature, interfering $H_2O$). A broader window may have 9 usable (i.e. non-saturated) $N_2O$ lines, but if the fits are 3x worse due to interferences, the computed retrieval uncertainty will be the same as the narrow window.

The figures below show averaging kernels for four $N_2O$ windows, with decreasing line intensities from left to right. In the upper panels the kernels are plotted as a function of altitude. In the lower panels the same kernels are plotted as a function of pressure. The kernels are computed for a representative subset of 140 ground-based spectra covering altitudes from 0 to 3.8 km, and temperatures from -40 to +40C. Kernels are color-coded by airmass, the most important
factor governing the shape. Red denotes an airmass of 9, orange 7, green 5, light blue 3, blue 2, and purple denotes an airmass of 1.



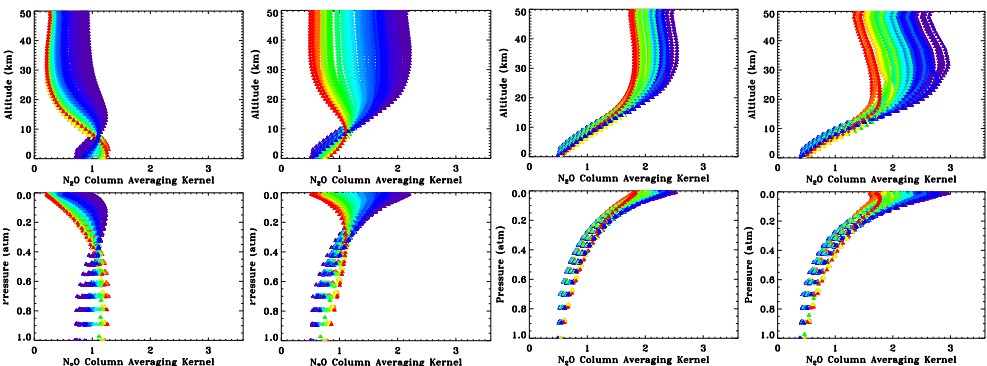

***Figure C.1.*** *Averaging kernels for four $N_2O$ windows of decreasing line strength left to right.*
*Top panels plot kernels versus altitude. Bottom panels plot same kernels versus pressure. Left-*
*most panels show results for 2539 $cm^{-1}$ window containing strong $N_2O$ lines. Next is the less*
*strong 2479 $cm^{-1}$ window. Next is the medium strength 3372 $cm^{-1}$ window. Far-Right panel show*
*kernels from 2781 $cm^{-1}$ window containing weak $N_2O$ lines.*

In summary, we rejected the 2806 window (#3) based on its poor CC, a likely

consequence its low E"_bar. We rejected the 2796 $cm^{-1}$ window (#9) because it currently

produces a high bias of 6% relative to the other windows, likely due to our neglect of line-mixing.

We rejected the 2539 and 2580 $cm^{-1}$ windows (# 6,7) because they are much stronger than the

others and therefore have much smaller averaging kernels in the stratosphere. The remaining

eleven $N_2O$ windows were averaged, after correcting for their biases, and used to create the

results presented in the remainder of the paper.