# Peer review of "Atmospheric Carbonyl Sulfide (OCS) measured remotely by FTIR solar absorption spectrometry"

_Atmospheric Chemistry and Physics, 2017_

## Referee Comment (RC1) · Anonymous Referee #1 · 3 Jul 2017

General comments: The authors present the measurements of OCS using MkIV FTIR spectrometer from both balloon campaigns and ground-based observations, and analyze the long-term trend and seasonal cycle. OCS is suggested to provide additional insights on carbon cycle, because of its similarity to CO2 during plant uptake. To use column measurements in the application, the OCS variations in the troposphere need to be extracted out. In this paper, the N2O column measurements are used to account/correct the stratospheric variations, because OCS and N2O share a similar profile shape and N2O is stable in the troposphere, which has been used on CH4 in other studies. This paper is a valuable contribution for making use of the OCS column measurements on the tropospheric variation. I recommend publication of this work in

[Figure]

ACP after minor revisions. Specific comments: 1. Line 51: it may worth to write the current uncertainties of using OCS to study the carbon cycle, such as the ocean and soil. It does not need to be a full review, but not mentioning it at all could not give the readers a clear view on the topic. 2. Line 125: could you explain more detail on why the weaker OCS bands provide more information than the strong bands at lower altitudes? Maybe show the AVKs from different bands. 3. line 190: Can authors give the confidence level of the relationship? It would be good to mention this uncertainty when using N2O2K to correct OCS stratospheric variations. 4. It would be better show the linear fitting between P and N2O in Fig.A.2, and mark the Pb and b. It will help the readers to understand how the N2O column above Pb is calculated in line 724. Technical corrections: 1. The format of the citations should be consistent, the authors sometimes use "()", sometimes use "[]". I think ACP uses "()". 2. Line 27: the full name of CS2 should go to the previous sentence where it's mentioned the first time. 3. line 116: Figure 1: the titles of subfigures are cut off; the y-axis of upper right panel is not clear. The same problem is also in the Figure 5. 4. Line 706: change "N2O=120 ppt" to "N2O=120 ppb".

---

## Referee Comment (RC2) · Anonymous Referee #2 · 2 Aug 2017

**Review of Toon et al: *Atmospheric Carbonyl Sulphide (OCS) measured remotely by FTIR solar absorption spectrometry* for Atmospheric Chemistry and Physics**

August 2, 2017

The article by Toon et al is part of a special issue celebrating twenty-five years of the Network for Detection of Atmospheric Composition Change (NDACC) between ACP, AMT and ESSD. The paper describes the retrieval and resulting timeseries of carbonyl sulfide (OCS) from solar absorption FTIR measurements using the MkIV spectrometer, covering that 25 year period.

Traditionally, interest in atmospheric abundances of OCS has centred around the fact that it is the most abundant sulfur gas in the atmosphere and its role as a precursor for the stratospheric sulfate aerosol layer. More recently, studies have explored the potential of using OCS as a tracer for photosynthetic uptake, which would allow some additional understanding of the partitioning of net ecosystem exchange of $CO_2$ between respiration and photosynthesis components, which are co-located and difficult to differentiate between via direct $CO_2$ flux measurements or inverse modelling from atmospheric concentration measurements. OCS measurements, however, are relatively rare in comparison to those of $CO_2$, so it is therefore encouraging to see publication of long timeseries of OCS amounts.

One interesting aspect is the removal of stratospheric variability using the relationship with $N_2O$, something that has also previously been utilised to look at tropospheric methane columns.

In general, the paper is scientifically sound and worthy of publication. I would suggest that it perhaps more suited for AMT than ACP, but given

this is a shared special issue and it is not clear cut, I won't insist on that!

I have two major comments, though I hesitate to use the word major.

1. throughout the paper, references to S are spelt with 'ph'. While this is acceptable in (British) English, the IUPAC spelling uses an 'f'. I believe that is what should be used here.

2. the paper could benefit from some careful reading and editing. It currently reads a lot like it was written in a rush to meet a deadline. There are lots of places where the language could be tighter or more formal. There are also places where it is a bit repetitive and/or long-winded. I will try to point out these, but no doubt I will miss some so I'd encourage a careful re-reading during the revision process.

After addressing these and other minor comments the manuscript will be suitable for publication.

**1 General Comments**

- There are a number of sentences beginning with "And" or "So" that could be rephrased.

- Figures using colour scales should have a legend/key with the colour scale on them rather than having to search for a description in the caption

**2 Technical Comments**

- line 32 result → results

- lines 38-42 - what about the relative affinity of plants towards $CO_2$ and OCS? How does this affect the seasonal cycles?

- line 44 - ATMOS has not been introduced yet, but is used as an acronym several times before it is defined (currently about line 97-98)

- line 57 - decrease → decreased

- line 59 - repeat of "OCS column"

- line 100 - the section could maybe be split somewhere or renamed, as it covers not only the observations but also details on the fitting.

- line 108/Table 1 - millions of cubic feet? Is this actually relevant? If so, should it not be in SI units? Otherwise omit.

- Figure 1 - I realise it would clutter the figure, but it would be nice to see the interfering gases plotted in the spectrum as well. Maybe offset vertically from the OCS.

- Figure 2 and elsewhere - it would be good to have a key for the colour scale in the figure somewhere, rather than just a description in the caption

- line 152 - exact same → same

- line 160 - maybe relate the green points to what this means w.r.t. time.

- line 160, 167 - "it is clear" - maybe tighten the language here - this is somewhat subjective.

- line 180 - incomplete sentence here.

- line 186 - "fairly" linear - what is the uncertainty in this fit? e.g. what is the residual around the linear fit in this range?

- line 216 then → than

- line 221 - Max → maximum

- line 222 - e.g. clouds - clarify?

- line 223 - why the different number of coadds at lower airmasses? How were these numbers chosen (i.e. whether to average four or six spectra)?

- line 224 - maybe "averaged" instead of "average"

- line 238 - given you have used different selections of windows between the balloon- and ground-based spectra, can you be certain of the consistency between the analyses?

- line 244 - I appreciate this is going into an NDACC special issue, but there is no prior context in this article for NDACC or how it operates within the infrared working group.

- line 255 - you could maybe refer to previous work where stratospheric relationships between species have been used to remove stratospheric variability or infer tropospheric abundances.

- line 257-259 - was the consistency of the AKs the major criterion used in the selection? It would be nice to be clear here.

- line 262 - the same? or just similar?

- Figure 4 - legend for colours would be nice

- line 281 - how are you judging the precision and accuracy of your OCS retrievals?

- line 292 - the interfering species could be shown offset to the OCS. It would be good to see these, particularly to observe if there is any relationship with the patterns in the residuals.

- line 300 - remove "of course"

- line 302 - how are the surface pressure and H2O column used to infer the dry air column?

- line 307 - delete "So" and maybe add "therefore after $N_2O$

- line 308 - is this really noise, or rather variability?

- line 309 - can you not quantify what the spread of values is?

- line 325 - season $\rightarrow$ seasonal

- line 329-330 - presumably, however, the fitting improves going from 2 harmonics to 3. Maybe clarify

- line 370 - was this the only time measuring at Fairbanks?

- line 372 - it is probably true that the enhanced seasonal cycle relative to other locations is due to the proximity of the boreal forest, but maybe you could explain why this is likely to increase the seasonal cycle amplitude

- line 407 - maybe briefly comment on how site-to-site biases could be assessed (or not, given the relative feasibility)

- Table 4 and elsewhere - what do you actual mean by precision? Precision is a term that can refer to a number of things, so a definition of how you are calculating/determining precision here would be good. Presumably it is something like the standard deviation in the retrievals over a period of time that you expect the atmospheric state to not change.

- line 428-429 - the large seasonal cycle reported in Griffith et al came from one year's data. Can the consistency of this with later years be inferred from any other studies (e.g. Kremser et al)?

- line 443 - wording at the end of the sentence

- line 451 - how was xCO2 retrieved?

- line 458-461 - maybe tighten the language and reporting of the analysis here

- line 472-473 - maybe reword to make this less subjective

- paragraph from line 485 - this seems like it warrants a separate section, or that it should be condensed.

- line 518 - base $\rightarrow$ based

- line 521-523 - this could maybe be mentioned earlier

- line 526 - rms is an acronym, so capitalise

- line 705 - 120 ppt $\rightarrow$ 120 ppb. Also consistency - is it linear above 100 or 120 ppb?

- line 752 and thereafter - these are not complete sentences, reword please.

- line 824 - it would be interesting to see what average residuals look like, but admittedly most likely going into too much detail for this manuscript

- Table B.2. - is colour necessary here? You could separate with bolded horizontal lines or alternating shaded and non-shaded

- line 875-876 - to play devil's advocate - if these windows are systematically lower, could you not just scale them to match the other windows? In fact, I guess it is not inconceivable that these are correct (though unlikely, I will concede), because we do not know the "truth" here.

- line 880 - the values do not exceed 1 by very much...

- line 888 - subjectivity here

- line 895 - the worst? Or just having the worst agreement with the other windows?

- line 899 - maybe formalise the language a little

- line 934 - spectral $\rightarrow$ spectra

- line 939-940 - maybe explicitly state why this requires favouring the weaker $N_2O$ lines (presumably because the OCS lines are weak).

- line 945 and thereafter - repetitive from Appendix B, so you could refer to that instead of explaining them again

- Table C.2. - units for center and width

- line 964 - you didn't define line mixing as LM

- line 969 - again - not necessarily the worst, but in the worst agreement with the other windows

- line 970 - formatting with E" etc.

- line 974 - why is this surprising? Unless I have misunderstood, you have argued extensively that the narrow windows should have better fits/RMS.

Apologies for the delayed response.

**3   References**

---

## Author Comment (AC1) · 5 Oct 2017

Reviewer comments in black.
Author responses in blue.

General comments:

The authors present the measurements of OCS using MkIV FTIR spectrometer from both balloon campaigns and ground-based observations, and analyze the long-term trend and seasonal cycle. OCS is suggested to provide additional insights on carbon cycle, because of its similarity to CO2 during plant uptake. To use column measurements in the application, the OCS variations in the troposphere need to be extracted out. In this paper, the N2O column measurements are used to account/correct the stratospheric variations, because OCS and N2O share a similar profile shape and N2O is stable in the troposphere, which has been used on CH4 in other studies. This paper is a valuable contribution for making use of the OCS column measurements on the tropospheric variation. I recommend publication of this work in ACPD ACP after minor revisions.
Thank you.

Specific comments:
1. Line 51: it may worth to write the current uncertainties of using OCS to study the carbon cycle, such as the ocean and soil. It does not need to be a full review, but not mentioning it at all could not give the readers a clear view on the topic.
Agreed. Added the sentences: " $CO_2$ measurements alone can only determine net biosphere flux, but cannot differentiate between photosynthesis and respiration. OCS is also taken up by plants during photosynthesis but is not respired, and so may be able to help distinguish between these processes (Wang et al., 2012).

2. Line 125: could you explain more detail on why the weaker OCS bands provide more information than the strong bands at lower altitudes? Maybe show the AVKs from different bands.
L125 doesn't say "more information", it says "additional information". At low tangent altitudes the strong OCS lines of the v3 band saturate and also become blacked out by strong interfering absorption by H2O and CO2. Look at the lower panels of figure 1. Blacked out OCS lines don't provide any information to the retrieval. The weaker OCS bands, however, are in less cluttered spectral regions and don't get so blacked out at the lower altitudes and therefore provide relatively more information to the retrieval.

3. Line 190: Can authors give the confidence level of the relationship? It would be good to mention this uncertainty when using N2O2K to correct OCS stratospheric variations.
Added the following sentence to the Fig.A.1 caption: " A straight line fitted to the $N_2O^{2K} > 120$ ppb data (417 points) has a gradient of 0.22489 +/- 0.00202, an intercept of $N_2O = 118.4$ +/- 0.8 ppb, and a Pearson correlation coefficient is 0.982. "

4. It would be better show the linear fitting between P and N2O in Fig.A.2, and mark the Pb and b.
The right-hand panel of Fig A.2 shows P and N2O. Are you suggesting dropping the left hand panel?

It will help the readers to understand how the N2O column above Pb is calculated in line 724.
Agreed. I have added dotted lines with Pb and b labeled to figure A.2.

Technical corrections:

1. The format of the citations should be consistent, the authors sometimes use "()", sometimes use "[]". I think ACP uses "()".
Agreed and done.

2. Line 27: the full name of CS2 should go to the previous sentence where it's mentioned the first time.
Done.

3. line 116: Figure 1: the titles of subfigures are cut off;
Yes, this is to stop it running into the next panel. I could completely remove the text at the top of each panel, but this loses information, like the zenith angle, tangent altitude, rms fit, etc. So I tried to crop it a bit more neatly.

the y-axis of upper right panel is not clear.
Are referring to the slight overlap of the y-axis annotation? If so, this has been fixed.

The same problem is also in the Figure 5.
I've tried to tidy it up by additional cropping, but it is still not perfect.

4. Line 706: change "N2O=120 ppt" to "N2O=120 ppb".
Fixed.

---

## Author Comment (AC2) · 5 Oct 2017

Review of Toon et al: Atmospheric Carbonyl Sulphide (OCS) measured remotely by FTIR solar absorption spectrometry for Atmospheric Chemistry and Physics August 2, 2017

Reviewer comments in black.
Author responses in blue.

The article by Toon et al is part of a special issue celebrating twenty- five years of the Network for Detection of Atmospheric Composition Change (NDACC) between ACP, AMT and ESSD. The paper describes the retrieval and resulting timeseries of carbonyl sulfide (OCS) from solar absorption FTIR measurements using the MkIV spectrometer, covering that 25 year period. Traditionally, interest in atmospheric abundances of OCS has centred around the fact that it is the most abundant sulfur gas in the atmosphere and its role as a precursor for the stratospheric sulfate aerosol layer. More recently, studies have explored the potential of using OCS as a tracer for photosynthetic uptake, which would allow some additional understanding of the partitioning of net ecosystem exchange of CO2 between respiration and photosynthesis components, which are co-located and difficult to differentiate between via direct CO2 flux measurements or inverse modelling from atmospheric concentration measurements. OCS measurements, however, are relatively rare in comparison to those of CO2, so it is therefore encouraging to see publication of long timeseries of OCS amounts. One interesting aspect is the removal of stratospheric variability using the relationship with N2O, something that has also previously been utilised to look at tropospheric methane columns. In general, the paper is scientifically sound and worthy of publication. I would suggest that it perhaps more suited for AMT than ACP, but given this is a shared special issue and it is not clear cut, I won't insist on that! I have two major comments, though I hesitate to use the word major.
Thank you for the kind comments and the very careful and detailed review.

1. throughout the paper, references to S are spelt with 'ph'. While this is acceptable in (British) English, the IUPAC spelling uses an 'f'. I believe that is what should be used here.
Okay, done. And for consistency, I have also changed sulphur to sulfur and sulphate to sulfate, but not in the references, where the original spelling is retained.

2. the paper could benefit from some careful reading and editing. It currently reads a lot like it was written in a rush to meet a deadline.
It certainly was.

There are lots of places where the language could be tighter or more formal.
Agreed.

There are also places where it is a bit repetitive and/or longwinded. I will try to point out these, but no doubt I will miss some so I'd encourage a careful re-reading during the revision process. After addressing these and other minor comments the manuscript will be suitable for publication.
Thank you.

1 General Comments
• There are a number of sentences beginning with "And" or "So" that could be rephrased.
Done.

• Figures using colour scales should have a legend/key with the colour scale on them rather than having to search for a description in the caption
I don't have an easy way of doing this.

2 Technical Comments
• line 32 result → results
Fixed

• lines 38-42 - what about the relative affinity of plants towards CO2 and OCS? How does this affect the seasonal cycles?

Added the sentence: "Plants have an equal affinity to $CO_2$ and OCS, in terms of their stomatal conductance and mesophyll diffusion, but the OCS has a ten times higher biochemical activity (Berry et al., 2013), which may vary with plant type."

How does this affect the seasonal cycles?
In fractional terms, OCS has a much larger seasonal cycle than CO2, as stated in lines 37-40. But I don't know whether this is directly attributable to its higher biochemical activity.

• line 44 - ATMOS has not been introduced yet, but is used as an acronym several times before it is defined (currently about line 97-98)
Fixed.

• line 57 - decrease → decreased
Fixed.

• line 59 - repeat of "OCS column" 2
Fixed.

• line 100 - the section could maybe be split somewhere or renamed, as it covers not only the observations but also details on the fitting.
Agreed. Have split this long section into Balloon Observations starting at line 100 and Balloon Results starting at line 163. Have also added section numbers.

• line 108/Table 1 - millions of cubic feet? Is this actually relevant? If so, should it not be in SI units? Otherwise omit.
Omitted the MCF column.

• Figure 1 - I realise it would clutter the figure, but it would be nice to see the interfering gases plotted in the spectrum as well. Maybe offset vertically from the OCS.
My first attempt at this figure did include all interfering gases, but visually it was a mess. Off-setting would help, but you have to offset a lot because these are wide windows and so the H2O and CO2 contributions go steeply from 0 to 1. Instead I have made an x-zoomed, fully-colored, version of fig.1 and will include it as Supplementary Information, since I think that relatively few readers would benefit from having these plots in the main paper. The new SI is 7 pages long and its caption states:
"***Figure S.1.*** *The following 6 pages show spectral fits to balloon spectra. The black diamond symbols represent the measured transmittances, and the black line is the fitted calculation. The colored lines are the contributions of the various fitted gases to the fitted calculation. The short panel at the top of each spectral fit shows the spectral fitting residuals (measured - calculated). This figure shows the same data as fig.1 of the main paper, but includes the contributions of the various interfering gases with an expanded x-scale, allowing them to be clearly distinguished. As in fig.1, the upper panels show fits to the 24.0 km tangent altitude spectrum, and the lower panels shows fits to the 8.6 km tangent altitude spectrum. The first four pages show consecutive sections of the 2038.1-2062.3 cm$^{-1}$ window, each about 6 cm$^{-1}$ wide. The last two pages show fits to the 2063.3-2076.0 cm$^{-1}$ window.*

• Figure 2 and elsewhere - it would be good to have a key for the colour scale in the figure somewhere, rather than just a description in the caption
You would need two color scales: one for the left hand panels and one for the right hand panels. While I agree that this would be nice, I don't have the time to investigate how to do this right now.

• line 152 - exact same → same
Done.

• line 160 - maybe relate the green points to what this means w.r.t. time.
Sentence now reads: "It is clear that the green points, measured 1997-2003 at high latitudes, have lower OCS amounts than the other flights."

• line 160, 167 - "it is clear" - maybe tighten the language here - this is somewhat subjective.
L160 now states: "The green points, measured 1997-2003 at high latitudes, have lower OCS amounts than the other flights."
L167 now states ".  Panel 2c shows that in the later years (red) there is more $N_2O$ at a given OCS value than in the early years (blue)."

• line 180 - incomplete sentence here.
Fixed.

• line 186 - "fairly" linear - what is the uncertainty in this fit? e.g. what is the residual around the linear fit in this range?
changed to " The OCS-$N_2O^{2K}$ relationship plotted in Fig. 2e is highly linear (Pearson correlation coefficient of 0.982) for $N_2O^{2K}$ values down to 120 ppb

Also, fig 2e is replicated in Appendix A, where the caption now states: ).  *A straight line fitted to the $N_2O^{2K}$ > 120 ppb data (417 points) has a gradient of 0.225 ± 0.002, an intercept of $N_2O$ =118.4 ± 0.8 ppb, and a Pearson correlation coefficient of 0.982.*"

• line 216 then → than
Fixed.

• line 221 - Max → maximum
Changed "Max OPD" to "maximum optical path difference"

• line 222 - e.g. clouds - clarify?
Changed to " when clouds blocked the sun"

• line 223 - why the different number of coadds at lower airmasses? How were these numbers chosen (i.e. whether to average four or six spectra)?
I could write a whole paragraph on this, but it would interrupt the flow of the discussion.
Sentence now states: "After discarding bad spectra (e.g., when clouds blocked the sun) the remainder are averaged into forward-reverse pairs at high solar zenith angles when the airmass is changing rapidly, or in fours or sixes at lower zenith angles when the airmass is changing slowly."

• line 224 - maybe "averaged" instead of "average"
I'm not sure.  Six **averaged** spectra result in one **average** spectrum.  In the sentence on L224 I'm referring to the output of the averaging process, not the input.

• line 238 - given you have used different selections of windows between the balloon- and ground-based spectra, can you be certain of the consistency between the analyses?
The selection of windows is not that different.  In both cases the OCS information is coming primarily from the v3 band.  Yes, the balloon windows are somewhat wider, but the self-consistency of the spectroscopy within a band should be excellent, so adding more lines from the same band shouldn't bias the retrieval (in the absence of large residuals).  Although the balloon retrievals use additional weaker OCS bands, Table 2 shows that the results from these weaker bands are collectively no different from the strong v3 band.

• line 244 - I appreciate this is going into an NDACC special issue, but there is no prior context in this article for NDACC or how it operates within the infrared working group.

I assumed that the special issue would have an overview paper at the beginning. But you are right, each individual paper should stand alone. I have therefore spelled out the NDACC acronym at first use, and have removed the IRWG acronym.

• line 255 - you could maybe refer to previous work where stratospheric relationships between species have been used to remove stratospheric variability or infer tropospheric abundances.

Added ". "Wang et al. (2014) used $N_2O$ in this manner to remove stratospheric variations from column $CH_4$.". Also added the Wang (2014) reference:

• line 257-259 - was the consistency of the AKs the major criterion used in the selection? It would be nice to be clear here.

Yes, it was a major criterion. To make this clearer, this sentence now states "Suffice it to say that a subset of $N_2O$ windows was selected based on three criteria: 1) high precision, 2) consistency in the retrieved $N_2O$ amounts, and 3) averaging kernels similar to those of OCS."

• line 262 - the same? or just similar?

Changed sentence to " Their closely matching shapes means that information about atmospheric OCS and $N_2O$ has a similar altitude distribution and is therefore directly comparable.

• Figure 4 - legend for colours would be nice

I don't have an easy way of doing this. I have expanded the caption to mention more colors.

• line 281 - how are you judging the precision and accuracy of your OCS retrievals?

The spectral fitting software computes an uncertainty on every single retrieved OCS values.

• line 292 - the interfering species could be shown offset to the OCS. It would be good to see these, particularly to observe if there is any relationship with the patterns in the residuals.

As with the balloon case, this was my initial inclination, to show all the interferers. But the resulting figure looked like a mess. In the ground-based case, in addition to OCS there were 10 interfering gases (2 CO isotopologs, 3 CO2 isotopologs, 2 H2O isotopologs, O3, solar, and Other), so color separation is an issue (we are forced to use similar colors that the eye can't distinguish). For this reason I'm reluctant to make another Supplementary Information, as I did for the balloon case. And yes, the patterns in the residuals are related to specific interferers, most notable H2O and CO2, as stated in the caption.

• line 300 - remove "of course"

Done.

• line 302 - how are the surface pressure and H2O column used to infer the dry air column?

Sentence now states: " Figs. 6c and 6d shows xOCS: the OCS column divided by the dry air column, the latter obtained by subtracting the H2O column from the total column of all gases, which is inferred from the measured surface pressure."

• line 307 - delete "So" and maybe add "therefore after N2O

Okay.

• line 308 - is this really noise, or rather variability?

Agreed. Sentence now states : ". We know from the balloon measurements that OCS and $N_2O$ have similarly-shaped vmr profiles (at least up to 30 km) and are both subject to the same dynamical perturbations, therefore dividing the OCS by $N_2O$ cancels most of the transport-driven variations."

• line 309 - can you not quantify what the spread of values is?
Yes, I did: 10%. I removed the word "probably"

• line 325 - season → seasonal
Fixed.

• line 329-330 - presumably, however, the fitting improves going from 2 harmonics to 3. Maybe clarify
Sentence now states: " The choice of 3 harmonics reflects the improved fit to the data as compared with 2 harmonics, but a lack of improvement with 4 harmonics.

• line 370 - was this the only time measuring at Fairbanks?
Yes. Sentence now states ". The blue data points, measured from Fairbanks, Alaska during the summer of 1997 (the only time we were there) show a much larger drawdown"

• line 372 - it is probably true that the enhanced seasonal cycle relative to other locations is due to the proximity of the boreal forest, but maybe you could explain why this is likely to increase the seasonal cycle amplitude
Sentence now states " This may be related to the location of Fairbanks within the boreal forest whose rapid summertime growth absorbs CO2 and OCS from the atmosphere. "

• line 407 - maybe briefly comment on how site-to-site biases could be assessed (or not, given the relative feasibility)
Well, I think that the analysis performed is a pretty good way of checking the consistency. To do better you would have to compare data from each site with output from a sophisticated model.

• Table 4 and elsewhere - what do you actual mean by precision? Precision is a term that can refer to a number of things, so a definition of how you are calculating/determining precision here would be good. Presumably it is something like the standard deviation in the retrievals over a period of time that you expect the atmospheric state to not change.
Exactly. Table 4 caption now has an additional sentence: "*The precision is the likely difference between observations made under nominally identical conditions.*"

• line 428-429 - the large seasonal cycle reported in Griffith et al came from one year's data. Can the consistency of this with later years be inferred from any other studies (e.g. Kremser et al)?
Good question. I believe that Kremser et al. were mainly focused on the long-term trends in OCS.

• line 443 - wording at the end of the sentence
Sentence now states " For the tropospheric partial column (3.6 - 8.9 km), they reported a 6% drop from 1995 to mid-2002, then a 7% increase to 2008, and flat since then."

• line 451 - how was xCO2 retrieved?
Sentence expanded to " Figure 7 shows the long-term secular changes and seasonal cycle of xCO2, derived from the same MkIV ground-based spectra using twenty $CO_2$ windows covering 2480 to 4924 cm$^{-1}$ (see http://mark4sun.jpl.nasa.gov/data/mkiv/all_mols_mir_1985_2016.gnd)"

• line 458-461 - maybe tighten the language and reporting of the analysis here
Modified sentence to: ". This similarity is consistent with OCS being absorbed by plants during photo-synthesis. Upon closer inspection of figure 7, the xCO2 peak occurs around day 130 and the fastest $xCO_2$ loss occurs around day 185. These are each about two weeks earlier than for xOCS."

• line 472-473 - maybe reword to make this less subjective
I believe that the offending sentence is: "These estimates do not quite overlap, but given that the altitudes and latitudes are different, the small discrepancy is not a cause for concern.", But I don'r

see anything subjective about it.

• paragraph from line 485 - this seems like it warrants a separate section, or that it should be condensed.
Deleted lines 502 - 506.

• line 518 - base → based
Fixed.

• line 521-523 - this could maybe be mentioned earlier
Inserted at line 219

• line 526 - rms is an acronym, so capitalise
Fixed here and 3 other places.

• line 705 - 120 ppt → 120 ppb.
Fixed.

Also consistency - is it linear above 100 or 120 ppb?
120. Fixed

• line 752 and thereafter - these are not complete sentences, reword please.
Changed to one long sentence: "In the latter category are the two windows used by Griffith et al. (1998) for analysis of spectra from Wollongong and Lauder, three windows used by Rinsland et al. (2002) for analysis of ground-based Kitt Peak spectra (2.1 km altitude) and subsequently used by Mahieu et al. (2003) for analysis of JFJ spectra, four windows used by Krysztofiak et al. (2015) for analysis of ground-based OCS from Paris, four windows used by Kremser et al. (2015) for analysis of three SH sites, and four windows used by Lejeune et al., (2017) for analysis of spectra from the Jungfraujoch (JFJ)."

• line 824 - it would be interesting to see what average residuals look like, but admittedly most likely going into too much detail for this manuscript
I agree, that would be interesting. But the residuals will likely look very different from the high-altitude sites than the low ones. And there were 21 windows evaluated. So that would be a lot of plots.

• Table B.2. - is colour necessary here? You could separate with bolded horizontal lines or alternating shaded and non-shaded
The colors in Table B.2 are not gratuitous. They match the colors at the top of fig. B.1, (a point that I neglected to mention) which I hoped would make it easier for the reader to relate the material between the table and figure. So perhaps this criticism is nullified goes away if I simply point out the common color scheme in the captions.

• line 875-876 - to play devil's advocate - if these windows are systematically lower, could you not just scale them to match the other windows?
It is probably not a simple scaling. For example, the bias might be H2O-dependent, or airmass-dependent.

In fact, I guess it is not inconceivable that these are correct (though unlikely, I will concede), because we do not know the "truth" here.
True. But the windows whose scale factors are furthest from 1.0 tend to have larger uncertainties in their scale factors and poorer correlation coefficients.

• line 880 - the values do not exceed 1 by very much...
True. But one has to draw the line somewhere.

• line 888 - subjectivity here
Changed " a pleasant surprise" to "surprising"

• line 895 - the worst? Or just having the worst agreement with the other windows?
Changed "worst" to "lowest"

• line 899 - maybe formalise the language a little
Changed "bad one" to "poor performer"

• line 934 - spectral → spectra
Fixed.

• line 939-940 - maybe explicitly state why this requires favouring the weaker N2O lines (presumably because the OCS lines are weak).
Yes. Sentence now states " But there is another consideration: we want to N2O averaging kernels to match those of OCS, which requires favoring the windows with the weaker N2O lines that match the depth of the OCS lines."

• line 945 and thereafter - repetitive from Appendix B, so you could refer to that instead of explaining them again
Agreed. Removed this sentence and added to Table C.2 caption: " The symbols $\Lambda_j$ , $\bar{\varepsilon}_j$ , $\chi^2/N$ are defined in appendix B"

• Table C.2. - units for center and width
Fixed.

• line 964 - you didn't define line mixing as LM
Fixed.

• line 969 - again - not necessarily the worst, but in the worst agreement with the other windows
Yes.  Two sentences earlier I had stated "A high PCC does not necessarily mean that a window is good. It just means that it consistently shows similar behavior to the mean of all windows.  ".

Changed "worst" to lowest"

• line 970 - formatting with E" etc.
Fixed.

• line 974 - why is this surprising? Unless I have misunderstood, you have argued extensively that the narrow windows should have better fits/RMS.
Deleted "It is perhaps surprising"